# Koopman-Assisted Trajectory Synthesis: A Data Augmentation Framework for Offline Imitation Learning

**Jin Wang, Pengcheng He, Ke Jiang, Xiaoyang Tan** *

College of Computer Science and Technology
Nanjing University of Aeronautics and Astronautics
MIIT Key Laboratory of Pattern Analysis and Machine Intelligence
Nanjing, JiangSu 210000, China
`{jinw1999,yukina233,ke_jiang,x.tan}@nuaa.edu.cn`

## Abstract

In offline imitation learning (IL), data augmentation is essential for addressing covariate shift, yet existing methods face a trade-off: single-step techniques risk dynamic inconsistency, while trajectory-level approaches suffer from compounding errors or scalability challenges. Recent Koopman-based methods, while promising, have been limited to single-step applications, creating computational burdens and vulnerabilities to approximation error. This paper introduces Koopman-Assisted Trajectory Synthesis (KATS), a framework designed to resolve these trade-offs. Our primary contribution is a novel trajectory-level synthesis process that inherently avoids the compounding errors of recursive rollouts. Second, we introduce a state-equivariant Koopman representation, a key innovation that ensures computational efficiency and scalability, unlike prior action-equivariant models. Third, we bolster robustness by integrating a refined generator matrix to counteract operator approximation errors. Through extensive experimentation, we demonstrate that KATS yields substantial improvements in policy performance and achieves new state-of-the-art results, proving especially effective in challenging, low-data-diversity regimes.

## 1 Introduction

Imitation Learning (IL) enables agents to acquire skills from expert demonstrations and has proven highly effective in fields like robotic control and autonomous driving Xie et al. (2024); Teng et al. (2022). A primary limitation of classic IL methods, such as behavioral cloning (BC), is their susceptibility to "covariate shift" Ross et al. (2011). This issue stems from a mismatch between the training distribution and the states visited during deployment, which triggers a compounding of errors and rapid performance decline Zare et al. (2024). The problem is especially severe in offline imitation learning, as the agent is confined to a fixed dataset and cannot interact with the environment to correct its deviations Prudencio et al. (2023).

The challenge of covariate shift is typically addressed through two distinct paradigms: algorithm-centric and data-centric methods. The former centers on enhancing algorithmic robustness, for example, by learning an explicit dynamics model to foresee future states Zhang et al. (2023) or by applying regularization to penalize actions in out-of-distribution (OOD) regions Chang et al. (2021). The latter, in contrast, focuses on improving the training data itself. Data augmentation stands out as a leading data-centric technique, as it offers a cost-effective means to enrich the dataset, enhance its diversity, and mitigate covariate shift without requiring further interaction with the environment Samadi et al. (2024).

The success of data augmentation hinges on generating high-quality, dynamically consistent data Kolev et al. (2024). Common single-step techniques like Mixup ignore temporal context, risking the creation of physically unrealistic states that can bias the model. Synthesizing entire trajectories is a

---

more robust approach, but it also faces hurdles. Forward-dynamics models suffer from compounding errors that lead to divergent trajectories, while powerful generative models like GANs Chen et al. (2024) or Diffusion models Janner et al. (2022) are often too data-hungry for limited offline datasets and struggle to generate novel yet plausible behaviors without a strong inductive bias.

This reveals the central need in offline IL for a framework that can generate novel, consistent trajectories without long-horizon error accumulation. The Koopman operator theory is uniquely suited to fill this gap. By transforming nonlinear dynamics into a linear evolution in a higher-dimensional space Mezić (2021), it simplifies control and analysis while also allowing for the easy integration of physical priors like symmetries. This feature directly enhances robustness with limited data, addressing a key failure point of other generative models. Consequently, Koopman-based data augmentation has garnered significant interest, spawning methods such as Koopman Q-learning (KFC) Weissenbacher et al. (2022), Koopman Model Predictive Control Wang et al. (2022), and Koopman linear quadratic regulators Shi & Meng (2022).

Existing Koopman-based methods, despite their potential, are constrained in their application to complex, high-dimensional control tasks Colbrook et al. (2024). A key issue stems from strategies that operate at the single-step transition level, such as the data augmentation in KFC Wang et al. (2022) Weissenbacher et al. (2022). By focusing on immediate state changes, these methods struggle to model the long-term temporal structures essential for coherent, multi-step task execution Efroni et al. (2021). This limitation highlights a clear need to explore the generation of full trajectories within the Koopman framework and to develop the associated theoretical guarantees for such an approach.

A second limitation is the potential for significant computational overhead. In the KFC method, for example, the reliance on an action-equivariance assumption creates a linear scaling relationship between the action dimension and both model complexity and inference cost, which constrains its scalability. A final, crucial weakness is the failure of these methods to properly address approximation errors from the Koopman linearization. Because the reliability of local linear approximations diminishes in areas with minimal or slow dynamics, the integrity of augmented data generated therein is compromised. The introduction of such unreliable data risks biasing the training process, which can guide the learned policy toward non-physical or unstable outcomes.

To overcome these challenges, we introduce Koopman-Assisted Trajectory Synthesis (KATS), a novel framework for offline imitation learning. Instead of traditional step-by-step rollouts, KATS generates new data by modeling entire expert trajectories in a learned latent space.

Our main contributions are as follows.

- We introduce a trajectory-level augmentation method that uses entire expert demonstrations as its base unit. This approach mitigates the compounding errors common in state-space rollouts and ensures generated trajectories are dynamically consistent within the linear Koopman space.

- Our framework leverages a state-equivariant assumption, which avoids the severe computational and memory costs of prior action-equivariant approaches. This design makes KATS highly efficient and scalable for complex tasks.

- We design a refined symmetric generator matrix that makes our model more robust to the inherent approximation errors of finite-dimensional Koopman representations, improving the quality of synthesized trajectories.

## 2 RELATED WORK

**Offline Imitation Learning and Distribution Shift.** A primary challenge in offline IL is distribution shift, where policies fail in out-of-distribution (OOD) states due to compounding errors Xu et al. (2020). Conservative methods address this by constraining the policy to the expert data distribution Kumar et al. (2020); Chang et al. (2021). Other approaches use learned dynamics models to plan back to known regions Shao et al. (2024); Janner et al. (2022) or explicitly estimate the data support to confine the policy Xu et al. (2022b). Our work complements these by focusing on data-centric mitigation through principled trajectory synthesis.

**Data Augmentation in Imitation Learning.** Data augmentation aims to alleviate distribution shift by expanding the dataset. While simple techniques like noise injection or random visual crops are common Laskey et al. (2017); Yang et al. (2022), they often produce dynamically inconsistent samples that can harm learning Gong et al. (2021). Unlike these methods, our work leverages Koopman theory's strong inductive bias for dynamics to generate entire trajectories that are both physically plausible and behaviorally consistent with the expert.

**Koopman Theory in Control and Reinforcement Learning.** Koopman operator theory enables the analysis of nonlinear systems via linear representations in a lifted space Mauroy et al. (2020), and has been widely used for learning dynamics Zhao et al. (2023), designing controllers Wang et al. (2022); Shi & Meng (2022), and policy learning in RL Weissenbacher et al. (2022). While some works have explored Koopman for data augmentation Jang et al. (2023), our method is distinct. We are the first to systematically develop a trajectory-level, state-equivariant Koopman augmentation framework specifically for offline IL. By integrating it with an Inverse Dynamics Model (IDM), we generate complete, dynamically consistent, and behaviorally plausible state-action sequences to tackle distribution shift.

## 3 PRELIMINARIES

This section provides the necessary background on offline imitation learning, focusing on the challenge of distribution shift and Koopman operator theory, detailing its definition and application in control systems.

### 3.1 OFFLINE IMITATION LEARNING AND DISTRIBUTION SHIFT

Imitation Learning (IL) aims to train a policy $\pi_\theta(a|s)$ to mimic an expert policy $\pi_E$ using a static dataset of expert demonstrations $\mathcal{D} = \{(s_i, a_i)\}_{i=1}^N$. A common approach, Behavioral Cloning, frames this as a supervised learning problem, minimizing a loss function such as the Mean Squared Error between the policy's actions and the expert's:

$$L_{BC}(\theta) = \frac{1}{N} \sum_{i=1}^{N} \|a_i - \pi_\theta(s_i)\|^2. \tag{1}$$

The primary challenge in this offline setting is distribution shift. The policy is trained on the expert's state distribution. During deployment, small errors can lead the agent to out-of-distribution (OOD) states not seen in $\mathcal{D}$. Since the policy behavior is undefined for these states and no new data can be collected, these errors compound, causing the agent's performance to degrade significantly.

### 3.2 KOOPMAN-BASED DATA AUGMENTATION

Koopman operator theory provides a powerful framework for linearizing nonlinear dynamics $s_{t+1} = F(s_t, a_t)$ in a high-dimensional lifted space. Recent methods, such as Koopman Forward Conservative Q-learning (KFC) Weissenbacher et al. (2022) Wang et al. (2022), leverage this theory for structured data augmentation, moving beyond simple noise injection.

In particular, the KFC method aims to identify a set of finite-dimensional observable functions, $z_t = g(s_t)$, that transform a system's state from its original domain to a higher-dimensional feature space. The objective of this transformation is twofold: to linearize the system's nonlinear dynamics and to ensure the resulting linear model is action-equivariant. In particular, this learned action-equivariant Koopman operator consists of matrices $(\mathbf{K}_0, \mathbf{K}_k)$ that approximate the system's dynamics in the lifted space as a bilinear model:

$$z_{t+1} \approx \left(\mathbf{K}_0 + \sum_{k=1}^{m} \mathbf{K}_k a_{t,k}\right) z_t. \tag{2}$$

where $z_t$ and $z_{t+1}$ are, respectively, two points in the lifted space, corresponding to a pair of states $(s_t, s_{t+1})$ in the original space.

Then, instead of adding arbitrary noise, KFC finds a symmetric generator matrix $\boldsymbol{\sigma}$ that represents dynamically consistent enhancements. This is achieved by solving the Sylvester equation, which imposes that the transformation commutes with the system dynamics. Let $\mathbf{K}_{a_t} \equiv \mathbf{K}_0 + \sum_{k=1}^{m} \mathbf{K}_k a_{t,k}$ denote the complete action-dependent Koopman operator for a given action $a_t$:

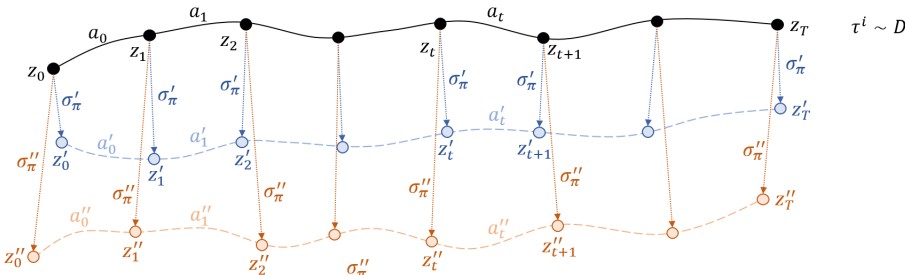

Figure 1: Overview of KATS framework: An Expert-Guided Koopman Model $K^{\pi_E}$ is learned from expert trajectories. Commuting transformations ($\sigma$) with $K^{\pi_E}$ and an inverse dynamics model generate augmented data to train policy $\pi_\theta$.

$$\mathbf{K}_{a_t}\boldsymbol{\sigma}_{a_t} - \boldsymbol{\sigma}_{a_t}\mathbf{K}_{a_t} = \mathbf{I}. \tag{3}$$

As illustrated in Figure 1 (a), to augment an original transition $(s_t, a_t, s_{t+1}, r_t)$, after embedding the states $s_t$ and $s_{t+1}$ into their high-dimensional latent representations, $z_t$ and $z_{t+1}$, the transformation $\boldsymbol{\sigma}_{a_t}$ is applied to generate augmented counterparts: $z_t^{a_t} = \boldsymbol{\sigma}_{a_t} z_t$ and $z_{t+1}^{a_t} = \boldsymbol{\sigma}_{a_t} z_{t+1}$. One can also use a decoder to project the generated new latent vectors back into the original state space, resulting in the augmented states $s_t^{a_t}$ and $s_{t+1}^{a_t}$. Finally, a complete, dynamically consistent transition tuple $(s_t^{a_t}, a_t, s_{t+1}^{a_t}, r_t)$ is synthesized by pairing these new states with the original action and reward Weissenbacher et al. (2022).

Despite its promise, the aforementioned *action-equivariant* Koopman-based data augmentation method has been limited to single-step applications. This restriction imposes significant computational burdens and introduces susceptibility to approximation errors. To address these challenges, we proposed *policy equivariant* Koopman-based data augmentation. In what follows, we first delve into the theoretical underpinnings of this and subsequently propose a practically viable solution.

## 4 KOOPMAN-ASSISTED TRAJECTORY GENERATION WITH POLICY-EQUIVARIANCE

In this section, we provide the theoretical foundation for our Koopman-Assisted Trajectory Synthesis (KATS) framework. Our central argument is that the symmetries of a closed-loop dynamical system, driven by a fixed expert policy, are directly reflected as commutation properties of its associated Koopman operator. We begin by defining this notion of symmetry, then establish a formal equivalence to an algebraic property of the operator, and finally show how this equivalence provides a rigorous mechanism for generating novel, policy-consistent trajectories.

**Definition 1** (Policy-Equivariant Dynamical System). *Consider the closed-loop dynamical system described by the discrete-time mapping $s_{t+1} = F_\pi(s_t)$, where $s_t \in \mathcal{S}$ is the state and $\pi : \mathcal{S} \to \mathcal{A}$ is a fixed policy. Let $\Sigma$ be a group with elements $\sigma : \mathcal{S} \to \mathcal{S}$ that act on the state space. The*

*system is called $\Sigma$-policy-equivariant if, for all $\sigma \in \Sigma$ and all $s \in \mathcal{S}$, the following condition holds: $F_\pi(\sigma \cdot s) = \sigma \cdot F_\pi(s)$. This implies that transforming the state and then applying the policy dynamics is equivalent to applying the dynamics first and then transforming the result.*

To analyze these potentially non-linear dynamics within a linear framework, we lift the system from the state space to a space of observables using the Koopman operator.

**Definition 2** (Koopman Operator for a Policy). *Let $\mathcal{K}(\mathcal{S}) = \{g : \mathcal{S} \to \mathbb{R}\}$ be the space of scalar-valued observable functions on the state space $\mathcal{S}$. The Koopman operator $K_\pi : \mathcal{K}(\mathcal{S}) \to \mathcal{K}(\mathcal{S})$ associated with the closed-loop system $F_\pi$ is a linear operator defined by: $[K_\pi g](s) \triangleq g(F_\pi(s))$. In a finite-dimensional approximation learned from data, $K_\pi$ is represented by a matrix that propagates a latent state vector $z_t = E(s_t)$ forward in time, i.e., $z_{t+1} = K_\pi z_t$.*

A key step in connecting the geometric state-space symmetry $\sigma$ to the algebraic operator $K_\pi$ is to define how $\sigma$ acts on the space of observables. This is achieved through an induced group action.

**Lemma 3** (Induced Group Action on Koopman Space). *A state space transformation $\sigma : \mathcal{S} \to \mathcal{S}$ induces a linear action on the space of observables $\mathcal{K}(\mathcal{S})$, defined by the pullback: $[\sigma g](s) \triangleq g(\sigma^{-1}(s))$.*

With these definitions in place, we can now state the central theorem of our analysis. It establishes a formal equivalence between the policy equivariance of the dynamics and a simple commutation relation involving the Koopman operator.

**Theorem 1** (Equivalence of Policy Equivariance and Koopman Commutation). *Let $F_\pi$ be a closed-loop dynamical system and $K_\pi$ be its corresponding Koopman operator. The following two statements are equivalent:*

1. *The system is $\Sigma$-policy-equivariant: $F_\pi(\sigma \cdot s) = \sigma \cdot F_\pi(s)$ for all $\sigma \in \Sigma, s \in \mathcal{S}$.*

2. *The Koopman operator commutes with the induced group action (where $\sigma$ acts on a function $g$ as $(\sigma g)(s) = g(\sigma^{-1}s)$): $K_\pi \sigma = \sigma K_\pi$ for all $\sigma \in \Sigma$.*

Theorem 1 is not merely a theoretical curiosity; it provides the direct mechanism for our data augmentation strategy. The following theorem makes this connection explicit, showing how the commutation property can be leveraged to generate new, valid system trajectories.

**Theorem 2** (Generating Policy-Consistent Trajectories via Symmetry). *Let $F_\pi$ be a $\Sigma$-policy-equivariant system with Koopman operator $K_\pi$ and let $\sigma \in \Sigma$. Let $\{z_t = E(s_t)\}$ be the latent representation of a trajectory under policy $\pi$, such that $z_{t+1} = K_\pi z_t$. Define a new trajectory $\{\hat{z}_t\}$ by $\hat{z}_t = \sigma \cdot z_t$. Then this new trajectory evolves according to the same Koopman dynamics: $\hat{z}_{t+1} = K_\pi \hat{z}_t$. Consequently, the decoded trajectory $\{\hat{s}_t = D(\hat{z}_t)\}$ is also a valid trajectory under the closed-loop dynamics of policy $\pi$.*

**Theorem 3** (Bounded Error for Transformed Trajectories). *Let $\{z_t^E\}$ be a latent expert trajectory with a sequence of one-step prediction errors $\{\epsilon_t\}$, where $\epsilon_t = z_{t+1}^E - \mathbf{K} z_t^E$. Let $\sigma$ be an expert-policy equivariant transformation with commutation error $\Delta$. Then the one-step prediction error $\hat{\epsilon}_t$ for the transformed trajectory $\{\hat{z}_t = \sigma(z_t^E)\}$ is given by:*

$$\hat{\epsilon}_t = \sigma(\epsilon_t) + \Delta z_t^E. \tag{4}$$

*Consequently, its norm is bounded as:*

$$\|\hat{\epsilon}_t\| \le \|\sigma\| \|\epsilon_t\| + \|\Delta\| \|z_t^E\|. \tag{5}$$

This result formally decomposes the error in a synthesized trajectory into two distinct sources:

1. **Propagated Model Error** ($\|\sigma\| \|\epsilon_t\|$)**:** This term reflects the inherent dynamics prediction error $\epsilon_t$, scaled by the magnitude of the transformation.

2. **Symmetry Violation Error** ($\|\Delta\| \|z_t^E\|$)**:** This term stems directly from the imperfect commutation between the learned symmetry and the dynamics model.

This decomposition provides a strong theoretical justification for our objective: it is crucial not only to minimize the standard prediction loss (reducing $\|\epsilon_t\|$) but also to explicitly regularize the model to

minimize the commutation residual $\|\Delta\|$. Please refer to Implication of the Bound in the Appendix for further discussion.

In summary, this analysis formally motivates our approach: by learning a Koopman operator $\mathbf{K}$ from expert data and then finding linear transformations $\sigma$ that satisfy the commutation relation $\mathbf{K}\sigma - \sigma\mathbf{K} \approx \mathbf{0}$, we are effectively identifying the underlying symmetries of the expert's behavior. Theorem 2 guarantees that applying these learned symmetries to existing trajectories will produce new data that is consistent with the expert's policy dynamics.

# 5 A Practical Algorithm

Motivated by the above analysis, in this section we detail our practical algorithm : Koopman-Assisted Trajectory Synthesis (KATS) , designed for augmenting datasets in trajectory-based offline imitation learning. The primary goal of KATS is to generate novel yet realistic data to improve policy learning. To this end, the framework synthesizes trajectories with three critical properties: dynamic consistency, behavioral plausibility, and computational efficiency. The KATS pipeline comprises three distinct stages: (1) trajectory synthesis via a behavior-driven Koopman operator, (2) robustness enhancement using an adaptive symmetry generation scheme, and (3) action inference through a decoupled inverse dynamics model. The entire process is visualized in Figure 1.

## 5.1 Koopman-Assisted Trajectory Synthesis

The first step of KATS is the construction of a generative model for expert trajectories. A key innovation of our method is the decision to model the closed-loop system dynamics governed by the expert policy, expressed as $s_{t+1} \approx G(s_t)$, as opposed to the conventional open-loop dynamics, $s_{t+1} = F(s_t, a_t)$. Consequently, the expert policy $\pi_E(a_t|s_t)$ becomes implicitly embedded within the transition function $G$. The practical implementation of this principle involves training an autoencoder ($E_\phi$, $D_\psi$) to map states $s$ into a latent space $\mathcal{Z}$ where the dynamics are linear. Within this linearized space, a single, action-independent Koopman operator $\mathbf{K}$ is then learned to propagate the system state:

$$z_{t+1} = E_\phi(s_{t+1}) \approx \mathbf{K}E_\phi(s_t) = \mathbf{K}z_t. \tag{6}$$

The model is trained by minimizing a combined loss function over the expert dataset $\mathcal{D}_E$, comprising state reconstruction and latent space prediction:

$$\mathcal{L}_{\text{recon}} = \mathbb{E}_{s \sim \mathcal{D}_E} \|s - D_\psi(E_\phi(s))\|^2, \tag{7}$$

$$\mathcal{L}_{\text{koopman}} = \mathbb{E}_{(s_t, s_{t+1}) \sim \mathcal{D}_E} \|E_\phi(s_{t+1}) - \mathbf{K}E_\phi(s_t)\|^2. \tag{8}$$

The proposed action-independent formulation offers two fundamental improvements upon prior methods such as KFC Weissenbacher et al. (2022). The first is a guarantee of behavioral plausibility: since the operator $\mathbf{K}$ is trained directly on expert closed-loop state transitions $(s_t, s_{t+1})$, it inherently embeds the expert's policy. Since the symmetry transformation commutes with the Koopman operator $\mathbf{K}$, the system dynamics are equivariant under this transformation. Consequently, applying the symmetry to a generated trajectory yields a new trajectory that remains consistent with the learned policy.The second improvement is a dramatic increase in efficiency. Our approach obviates the need for per-action modeling by learning only a single operator $\mathbf{K}$ and its corresponding symmetry basis $\sigma_j$ in a single offline step. This not only reduces storage complexity from $\mathcal{O}(N \cdot d^2)$ to a constant but also entirely removes the computational bottleneck associated with per-action processing.

## 5.2 Adaptive Symmetry Generation for Robustness

Any model-based approach is subject to its own approximation errors. A Koopman model may be inaccurate in regions of the state space where expert data is sparse. Naively generating data in such regions with a flawed model can inject noise and degrade policy performance.

To mitigate this, KATS introduces an adaptive data augmentation scheme. The core idea is to prioritize data synthesis in regions where the model is most uncertain. Instead of merely finding a fixed basis by solving the homogeneous Sylvester equation ($\mathbf{K}\sigma - \sigma\mathbf{K} = \mathbf{0}$), we learn the transformation $\sigma$ through an optimization process weighted by the Koopman model's prediction error. Specifically, the loss for $\sigma$ is defined as:

$$\mathcal{L}_\sigma = \mathbb{E}_{(s_t, s_{t+1}) \sim \mathcal{D}_E} w_{s_t} \|\sigma(z_{t+1}) - \mathbf{K}\sigma(z_t)\|^2, \tag{9}$$

where $z_t = E_\phi(s_t)$, and the weight $w_{s_t}$ is explicitly tied to the model's local error:

$$w_{s_t} = \exp(\tau \|z_{t+1} - \mathbf{K} z_t\|). \tag{10}$$

Here, $\tau$ is a temperature hyperparameter. This weighting scheme forces the learning of $\sigma$ to more strongly satisfy the symmetry property in areas where the original operator $\mathbf{K}$ struggles. In this way, KATS adaptively generates samples where they are most needed, turning a potential weakness into a targeted exploration strategy.

### 5.3 DECOUPLED ACTION INFERENCE VIA INVERSE DYNAMICS

Our action-independent modeling approach creates a new requirement: the generated symmetric state pairs $(s'_t, s'_{t+1})$ lack corresponding actions. To address this, we decouple action inference from dynamics modeling by training a separate Inverse Dynamics Model (IDM), $f_{\text{IDM}} : \mathcal{S} \times \mathcal{S} \to \mathcal{A}$. The IDM learns the mapping from state transitions to actions directly from the expert data:

$$\mathcal{L}_{\text{IDM}} = \mathbb{E}_{(s_t, a_t, s_{t+1}) \sim \mathcal{D}_E} \|a_t - f_{\text{IDM}}(s_t, s_{t+1})\|^2. \tag{11}$$

Once trained, this model infers a plausible action $a'_t = f_{\text{IDM}}(s'_t, s'_{t+1})$ for each generated state pair, creating a complete augmented tuple $(s'_t, a'_t, s'_{t+1})$.

### 5.4 IMPLEMENTATION DETAILS

The complete KATS pipeline is summarized in Algorithm 1. In the final step, we combine the original expert dataset $\mathcal{D}_E$ with our generated data $\mathcal{D}_{\text{aug}}$ to form a comprehensive training set, $\mathcal{D}_{\text{final}} = \mathcal{D}_E \cup \mathcal{D}_{\text{aug}}$. An imitation learning policy $\pi_\theta(a|s)$ is then trained on this enriched dataset, typically via behavioral cloning, to learn a more robust and generalizable policy:

$$\mathcal{L}_{\text{BC}}(\theta) = -\mathbb{E}_{(s,a) \sim \mathcal{D}_{\text{final}}} [\log \pi_\theta(a|s)]. \tag{12}$$

---

**Algorithm 1** Koopman-Assisted Trajectory Synthesis

---

**Require:** Expert trajectories $\mathcal{D}_E = \{\tau_i\}$, Koopman models $(E_\phi, D_\psi, \mathbf{K}_\pi)$ trained via Eq. 7 and Eq. 8, IDM $f_{\text{IDM}}$ trained via Eq. 11.
**Ensure:** Augmented dataset of trajectories $\mathcal{D}_{\text{aug}}$.
 1: Initialize $\mathcal{D}_{\text{aug}} \leftarrow \mathcal{D}_E$.
 2: Learn symmetry operators $\{\sigma_j\}_{j=1}^M$ satisfying $\mathbf{K}_\pi \sigma \approx \sigma \mathbf{K}_\pi$ via the objective in Eq. 9.
 3: **for** each expert trajectory $\tau = (s_0, \ldots, s_T)$ in $\mathcal{D}_E$ **do**
 4:   **for** each symmetry $\sigma_j$ in the basis **do**
 5:     Define the end-to-end state transformation $G_j(s) := (D_\psi \circ \sigma_j \circ E_\phi)(s)$.
 6:     Generate a new state trajectory:
 7:     $(s'_0, \ldots, s'_T) \leftarrow (G_j(s_0), \ldots, G_j(s_T))$.
 8:     Synthesize the corresponding action trajectory:
 9:     $(a'_0, \ldots, a'_{T-1}) \leftarrow (f_{\text{IDM}}(s'_0, s'_1), \ldots, f_{\text{IDM}}(s'_{T-1}, s'_T))$.
10:     Assemble and add the new trajectory $\tau' = (s'_0, a'_0, \ldots, s'_T)$ to $\mathcal{D}_{\text{aug}}$.
11:   **end for**
12: **end for**
13: **return** $\mathcal{D}_{\text{aug}}$.

---

## 6 EXPERIMENTS

In this section, we conduct a comprehensive empirical evaluation of our proposed method KATS. Our experiments are designed to rigorously assess its effectiveness, versatility, and robustness. We use the challenging D4RL benchmark Fu et al. (2020), which includes diverse locomotion (Mujoco), navigation (Antmaze) and manipulation (Adroit) tasks, serving as a standard to evaluate offline learning algorithms. All baseline algorithms are configured following the hyperparameters specified in their original publications to ensure a fair comparison.

Table 1: Comparative performance on custom control tasks. Best results for each task are in **bold**.

| Domain | Task Name | BC | SRA | MILO | KFC+BC | KATS+BC |
|--------|-----------|-----|-----|------|--------|---------|
| AntMaze | antmaze-umaze | $74.0 \pm 1.2$ | $85.3 \pm 1.1$ | $80.1 \pm 1.5$ | $79.1 \pm 3.4$ | $\mathbf{96.9 \pm 0.8}$ |
| | antmaze-umaze-diverse | $64.0 \pm 2.0$ | $81.5 \pm 1.8$ | $73.2 \pm 2.1$ | $66.2 \pm 1.5$ | $\mathbf{90.1 \pm 0.7}$ |
| | antmaze-medium-play | $68.2 \pm 3.1$ | $78.5 \pm 2.5$ | $67.5 \pm 3.3$ | $72.3 \pm 2.5$ | $\mathbf{82.7 \pm 1.4}$ |
| | antmaze-medium-diverse | $53.7 \pm 4.5$ | $60.2 \pm 3.9$ | $61.0 \pm 2.8$ | $57.1 \pm 2.2$ | $\mathbf{67.3 \pm 2.5}$ |
| | antmaze-large-play | $35.8 \pm 2.2$ | $48.3 \pm 2.0$ | $40.1 \pm 1.9$ | $42.8 \pm 1.5$ | $\mathbf{59.3 \pm 1.5}$ |
| | antmaze-large-diverse | $24.9 \pm 1.8$ | $39.5 \pm 1.5$ | $33.8 \pm 2.0$ | $28.1 \pm 1.7$ | $\mathbf{44.2 \pm 2.7}$ |
| Gym | halfcheetah-medium-expert | $55.2 \pm 1.8$ | $63.4 \pm 3.5$ | $44.5 \pm 1.5$ | $60.9 \pm 1.0$ | $\mathbf{81.2 \pm 6.7}$ |
| | walker-medium-expert | $107.5 \pm 2.0$ | $104.1 \pm 4.8$ | $95.4 \pm 3.8$ | $100.4 \pm 1.9$ | $\mathbf{110.4 \pm 4.6}$ |
| | hopper-medium-expert | $52.5 \pm 5.0$ | $104.5 \pm 3.3$ | $90.9 \pm 5.4$ | $70.2 \pm 1.3$ | $\mathbf{112.7 \pm 3.6}$ |
| Adroit | pen-human | $37.5 \pm 2.2$ | $40.2 \pm 2.0$ | $44.4 \pm 1.8$ | $41.3 \pm 1.5$ | $\mathbf{69.2 \pm 3.9}$ |
| | pen-cloned | $39.2 \pm 2.5$ | $45.8 \pm 2.3$ | $57.1 \pm 2.0$ | $38.3 \pm 1.7$ | $\mathbf{81.3 \pm 4.7}$ |
| | hammer-human | $4.4 \pm 0.8$ | $5.0 \pm 0.7$ | $5.9 \pm 0.6$ | $4.6 \pm 0.5$ | $\mathbf{7.2 \pm 0.3}$ |
| | hammer-cloned | $2.1 \pm 0.5$ | $2.3 \pm 0.4$ | $2.7 \pm 0.4$ | $2.8 \pm 0.3$ | $\mathbf{4.2 \pm 0.5}$ |
| | door-human | $9.9 \pm 1.5$ | $15.3 \pm 1.3$ | $27.0 \pm 1.0$ | $14.1 \pm 0.8$ | $\mathbf{37.2 \pm 2.1}$ |
| | door-cloned | $0.4 \pm 0.2$ | $0.6 \pm 0.2$ | $2.1 \pm 0.3$ | $1.6 \pm 0.3$ | $\mathbf{4.9 \pm 0.6}$ |

## 6.1 MAIN RESULTS

We benchmark KATS against a suite of prominent offline imitation learning algorithms: **Behavior Cloning (BC):** The fundamental baseline for imitation learning; **SRA** Shao et al. (2024): A state-of-the-art method that employs model-based reverse augmentation; **MILO** Chang et al. (2021): A strong model-based algorithm employing policy constraints; **KFC+BC** Weissenbacher et al. (2022): A baseline that combines Behavioral Cloning (BC) with data augmentation from KFC. Our experimental evaluation focuses specifically on D4RL datasets characterized by data sparsity and a limited number of expert demonstrations, as these scenarios pose the greatest challenge to offline learning algorithms.

Table 1 gives the comparative results for the task of offline IL, which unequivocally demonstrate the superiority of our proposed method, KATS. Across all 15 challenging control tasks, KATS+BC significantly outperforming all baselines, including vanilla Behavioral Cloning (BC), prior augmentation methods (SRA, MILO), and its direct predecessor, KFC+BC. The most critical insight comes from the direct comparison with KFC+BC. KATS+BC achieves massive performance gains in every single task, such as improving the score from 68.0 to 96.9 on antmaze-umaze. This significant improvement validates our core contribution: the action-independent formulation. By modeling the system's closed-loop dynamics, KATS effectively captures the expert's policy. Our empirical results suggest that this approach yields trajectories that are more behaviorally consistent than those from action-conditioned models like KFC.

Figure 2 gives some illustration of trajectories in the 'maze2d' environment. The visualizations reveal that our method generates paths with greater symmetry at corners and produces a more diverse set of samples in difficult areas, enriching the training data where it is most needed.

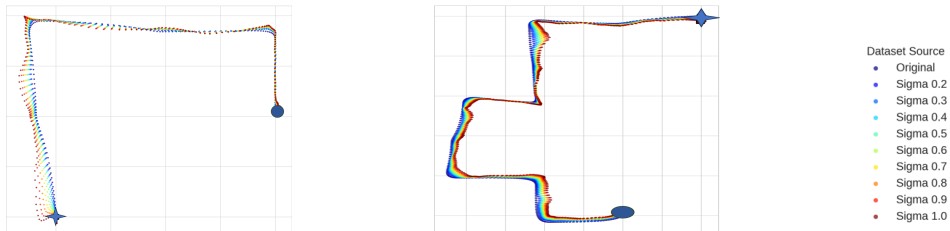

Figure 2: Ablation on the Blend Ratio of Symmetric Trajectories in `maze2d`.

Furthermore, the robust performance of KATS on sparse-reward navigation tasks (AntMaze) and noisy, suboptimal human demonstration data (Adroit) highlights its versatility and practical utility. It effectively extracts and amplifies the core skills from limited data, a critical capability for real-world

Table 2: Performance comparison against offline RL baselines on few-shot D4RL datasets. The best result in each row is highlighted in **bold**.

| Task | Size (ratio) | BC | CQL | IQL | DOGE | POR | TELS | Diffuser | KFC+CQL | KATS+BC |
|------|-------------|-----|-----|-----|------|-----|------|----------|---------|---------|
| Hopper-e | 10k (1%) | 69.1±12.4 | 72.4±6.1 | 78.5±9.3 | 83.4±17.9 | 87.4±18.0 | N/A | 74.5±3.8 | 89.1±7.6 | **97.1±10.4** |
| Hopper-me | 10k (0.5%) | 48.2±6.7 | 29.9±4.5 | 34.3±8.7 | 50.5±25.2 | 37.9±6.1 | **100.9±6.8** | 70.1±6.5 | 88.2±11.3 | 87.2±16.5 |
| Halfcheetah-e | 10k (1%) | 63.4±12.1 | 73.4±7.6 | 57.2±17.1 | 64.7±17.1 | 86.2±5.2 | N/A | 68.9±4.1 | 83.4±9.7 | **102.7±8.1** |
| Halfcheetah-me | 10k (0.5%) | 41.4±9.2 | 26.5±10.8 | 10.5±8.8 | 26.7±6.6 | 34.7±2.6 | 40.7±1.2 | 63.2±8.8 | 73.4±8.9 | **75.2±10.4** |
| Walker2d-e | 10k (1%) | 76.4±17.2 | 73.2±16.1 | 81.2±11.1 | 65.9±8.4 | 77.1±15.1 | N/A | 83.3±6.7 | 86.4±12.2 | **109.4±11.2** |
| Walker2d-me | 10k (0.5%) | 50.9±10.3 | 19.1±14.4 | 26.5±8.6 | 35.3±11.6 | 20.1±8.6 | 87.4±13.3 | 74.2±9.0 | 81.9±6.9 | **92.5±8.1** |
| Antmaze-u-d | 0.1M (10%) | 47.1±19.8 | 0.5±0.1 | 34.6±18.3 | 41.7±18.9 | 42.1±14.2 | 60.9±16.9 | 59.1±11.7 | 62.1±7.4 | **67.5±10.3** |
| Antmaze-u | 0.1M (10%) | 62.3±27.1 | 0.1±0.0 | 65.1±19.4 | 56.3±24.4 | 6.1±7.3 | **88.7±7.7** | 62.1±9.4 | 73.1±7.1 | 83.1±7.2 |
| Antmaze-m-d | 0.1M (10%) | 36.7±20.5 | N/A | 30.1±12.5 | N/A | N/A | 55.2±14.2 | 51.4±8.7 | 56.7±10.1 | **60.3±13.8** |
| Antmaze-m-p | 0.1M (10%) | 47.5±16.9 | N/A | 12.5±5.4 | N/A | N/A | 62.9±17.8 | 60.9±12.6 | 67.5±9.3 | **71.3±15.2** |
| Antmaze-l-d | 0.1M (10%) | 25.1±12.6 | N/A | 3.6±4.1 | N/A | N/A | 39.8±14.1 | 38.5±10.9 | 35.1±10.4 | **42.8±11.9** |
| Antmaze-l-p | 0.1M (10%) | 40.1±15.7 | N/A | 3.5±4.1 | N/A | N/A | 47.3±13.1 | 47.6±8.3 | 49.1±8.1 | **51.4±13.1** |

applications. In summary, these results confirm that KATS is a highly effective and general-purpose data augmentation technique that significantly pushes the frontier of offline imitation learning.

## 6.2 Results on the offline RL tasks

We also validate our method's efficacy in the offline reinforcement learning (RL) setting, which requires assigning rewards to augmented data. Adopting the strategy from KFC, we assign the reward for each synthetic transition by reusing the value from the corresponding step in the source demonstration. Our empirical evaluation includes a direct comparison with competitive offline RL algorithms: TELS Cheng et al. (2025), which employs temporal inverse dynamics regularization; POR Xu et al. (2022a), which leverages state-value guidance and an inverse dynamics model; and DOGE Li et al. (2022), which focuses on generalizing offline RL by exploiting data geometry.

Table 2 gives the results. One can see from the Table that KATS demonstrates remarkable efficacy in the challenging few-shot offline RL setting, achieving state-of-the-art results on 10 of the 12 tasks. Our results reveal a crucial insight: augmenting the dataset with high-fidelity, behaviorally consistent trajectories enables a simple Behavioral Cloning (BC) policy to decisively outperform complex algorithms designed to handle distributional shift.

Notably, these outstanding results were achieved using a simple reward assignment heuristic: reusing rewards from the source trajectory. While theoretically suboptimal, the success of this strategy strongly implies that the primary driver of performance is the high behavioral fidelity of the generated state-action trajectories. This finding is significant, as it shows that a superior generative model of behavior can allow a simple learning algorithm to surpass complex methods designed for value estimation and policy constraint in offline RL.

In conclusion, the results in Table 2 robustly demonstrate that KATS is not merely an imitation learning technique but a powerful and versatile data augmentation framework that effectively addresses the challenges of offline RL, especially in the critical and practical context of data scarcity.

## 6.3 Compatibility as a Plug-and-Play Augmentor

A key hypothesis is that KATS's data augmentation is model-agnostic and can benefit any offline IL algorithm. To test this, we treat KATS as a pre-processing step and apply it to the data sets used by BC and SRA Shao et al. (2024).

For baseline (BC, SRA), we compare its performance when trained on the original D4RL dataset versus being trained on the dataset augmented by KATS. We denote the augmented versions as 'BC + KATS' and 'SRA + KATS'.

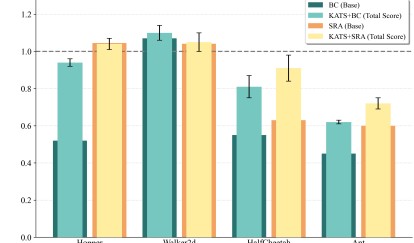

Figure 3: Performance comparison on offline IL benchmarks with KATS augmentation.

As shown in Figure 3, applying the KATS augmentation provides a significant boost in performance for BC and SRA. The improvement is particularly substantial for BC, which suffers from covariate shift, a problem that is mitigated by the

richer, symmetrically complete data. The fact that KATS enhances even a strong baseline like SRA validates its utility as a plug-and-play general-purpose module to improve data efficiency in offline imitation learning.

## 6.4 ABLATION STUDY ON COMPONENT EFFICACY.

To rigorously validate our design choices, we conducted an ablation study on the `antmaze` and `maze2d` benchmarks (Table 3). We compared the full **KATS** model against three variants:

- **KATS-$\sigma_A$**, which replaces the learnable $\sigma_\theta$-network with a fixed analytical matrix;
- **KATS w/o IDM**, which removes the Inverse Dynamics Model to test the necessity of action correction;
- **KFC+BC**, which employs an action-coupled encoder instead of our decoupled design.

As shown in Table 3, the full KATS model consistently outperforms all variants. The significant performance drop in *w/o IDM* confirms that the IDM is critical for resolving state-action mismatches in augmented trajectories. Furthermore, KATS surpasses *KATS-$\sigma_A$*, demonstrating that the adaptive $\sigma_\theta$-network captures more effective symmetry patterns than rigid analytical solutions.

Table 3: Ablation study on `antmaze` and `maze2d`.

| Domain | Task Name | BC | KFC+BC | KATS w/o IDM | KATS-$\sigma_A$ | KATS (Ours) |
|---|---|---|---|---|---|---|
| AntMaze | antmaze-umaze | $74.0 \pm 1.2$ | $79.1 \pm 3.4$ | $78.4 \pm 4.8$ | $87.3 \pm 2.6$ | $\mathbf{96.9 \pm 0.8}$ |
| | antmaze-umaze-diverse | $64.0 \pm 2.0$ | $66.2 \pm 1.5$ | $72.3 \pm 3.6$ | $82.1 \pm 3.3$ | $\mathbf{90.1 \pm 0.7}$ |
| | antmaze-medium-play | $68.2 \pm 3.1$ | $72.3 \pm 2.5$ | $69.5 \pm 2.8$ | $78.4 \pm 2.1$ | $\mathbf{82.7 \pm 1.4}$ |
| | antmaze-medium-diverse | $53.7 \pm 4.5$ | $57.1 \pm 2.2$ | $58.2 \pm 6.8$ | $65.2 \pm 4.5$ | $\mathbf{67.3 \pm 2.5}$ |
| | antmaze-large-play | $35.8 \pm 2.2$ | $42.8 \pm 1.5$ | $40.2 \pm 4.3$ | $51.7 \pm 4.0$ | $\mathbf{59.3 \pm 1.5}$ |
| | antmaze-large-diverse | $24.9 \pm 1.8$ | $28.1 \pm 1.7$ | $27.5 \pm 3.1$ | $36.7 \pm 2.6$ | $\mathbf{44.2 \pm 2.7}$ |
| Maze2d | maze2d-umaze | $72.1 \pm 6.0$ | $89.2 \pm 4.8$ | $86.6 \pm 8.1$ | $100.2 \pm 6.7$ | $\mathbf{113.2 \pm 5.6}$ |
| | maze2d-medium | $42.3 \pm 9.9$ | $63.7 \pm 8.5$ | $66.3 \pm 12.1$ | $90.8 \pm 11.5$ | $\mathbf{108.7 \pm 4.9}$ |
| | maze2d-large | $16.3 \pm 7.3$ | $55.9 \pm 9.1$ | $41.5 \pm 15.8$ | $89.2 \pm 12.2$ | $\mathbf{100.1 \pm 7.2}$ |

## 7 CONCLUSION AND FUTURE WORK

This paper introduced KATS, a novel data augmentation framework that leverages Koopman theory to address the critical distribution shift problem in offline imitation learning. By identifying and applying latent symmetries of the system dynamics, KATS generates diverse, yet dynamically consistent data. Our experiments validate that this principled approach effectively enhances the robustness and generalization of the learned policy against challenging out-of-distribution states.

**Limitations.** One limitation of the proposed method is that currently a relatively simple Koopman operator is learnt from data, hence it is interesting to explore more sophisticated Koopman architectures to capture highly nonlinear dynamics and integrating uncertainty principles.

**Acknowledgements** This work is partially supported by National Natural Science Foundation of China (62476128) and National Key R&D program of China (2021ZD0113203).

**Reproducibility.** We commit to releasing our core source code upon publication and provide it in the supplementary material for review. Our implementation uses PyTorch. All model architectures, hyperparameters, and training procedures are detailed in Appendix D. All results are averaged over 5 random seeds.

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

# A APPENDIX

This appendix provides the formal proofs and justifications for the definitions, lemmas, and theorems presented in Section 4. We restate each proposition for clarity before presenting its proof.

## A.1 JUSTIFICATION FOR LEMMA 3 (INDUCED GROUP ACTION)

**Lemma 4.** *A state space transformation $\sigma : \mathcal{S} \to \mathcal{S}$ induces a linear action on the space of observables $\mathcal{K}(\mathcal{S})$, defined by the pullback: $[\sigma g](s) \triangleq g(\sigma^{-1}(s))$.*

*Justification.* This definition, known as the pullback of the function $g$ by the map $\sigma$, is the standard way a transformation on a space induces a transformation on functions defined on that space. The intuition is as follows: the new function, $\sigma g$, when evaluated at a point $s$, should have the value that the original function $g$ had at the point that was mapped to $s$. If we denote this original point by $s'$, then $\sigma(s') = s$, which implies $s' = \sigma^{-1}(s)$. Therefore, the value of the new function at $s$ must be the value of the old function at $\sigma^{-1}(s)$, leading directly to the definition $[\sigma g](s) = g(\sigma^{-1}(s))$.

The linearity of this induced action can be shown directly. For any scalars $a, b \in \mathbb{R}$ and observables $g_1, g_2 \in \mathcal{K}(\mathcal{S})$:

$$
\begin{aligned}
[\sigma(ag_1 + bg_2)](s) &= (ag_1 + bg_2)(\sigma^{-1}(s)) \\
&= ag_1(\sigma^{-1}(s)) + bg_2(\sigma^{-1}(s)) \\
&= a[\sigma g_1](s) + b[\sigma g_2](s) \\
&= [a(\sigma g_1) + b(\sigma g_2)](s).
\end{aligned}
$$

Since this holds for all $s \in \mathcal{S}$, we have $\sigma(ag_1 + bg_2) = a(\sigma g_1) + b(\sigma g_2)$. $\square$

## A.2 PROOF OF THEOREM 1 (EQUIVALENCE OF POLICY EQUIVARIANCE AND KOOPMAN COMMUTATION)

**Theorem 4.** *Let $F_\pi$ be a closed-loop dynamical system and $K_\pi$ be its corresponding Koopman operator. The following two statements are equivalent:*

1. *The system is $\Sigma$-policy-equivariant: $F_\pi(\sigma \cdot s) = \sigma \cdot F_\pi(s)$ for all $\sigma \in \Sigma, s \in \mathcal{S}$.*

2. *The Koopman operator commutes with the induced group action: $K_\pi \sigma = \sigma K_\pi$ for all $\sigma \in \Sigma$.*

*Proof.* We prove both implications separately.

**Part 1: Equivariance implies Commutation ($1 \Rightarrow 2$).** Assume the system is $\Sigma$-policy-equivariant, i.e., $F_\pi(\sigma \cdot s) = \sigma \cdot F_\pi(s)$. We must show that $K_\pi \sigma = \sigma K_\pi$. To do this, we apply the operator $K_\pi \sigma$ to an arbitrary observable $g \in \mathcal{K}(\mathcal{S})$ and evaluate it at an arbitrary state $s \in \mathcal{S}$.

$$
\begin{aligned}
[K_\pi(\sigma g)](s) &= (\sigma g)(F_\pi(s)) && \text{(by Def. 2 of } K_\pi) \\
&= g(\sigma^{-1}(F_\pi(s))) && \text{(by Def. 3 of induced action)} \\
&= g(F_\pi(\sigma^{-1}(s))) && \text{(by policy equivariance, replacing } s \text{ with } \sigma^{-1}(s)) \\
&= [K_\pi g](\sigma^{-1}(s)) && \text{(by Def. 2 of } K_\pi) \\
&= [\sigma(K_\pi g)](s) && \text{(by Def. 3 of induced action)}
\end{aligned}
$$

Since $[K_\pi(\sigma g)](s) = [\sigma(K_\pi g)](s)$ for all $g \in \mathcal{K}(\mathcal{S})$ and all $s \in \mathcal{S}$, we conclude that the operators are equal: $K_\pi \sigma = \sigma K_\pi$.

**Part 2: Commutation implies Equivariance ($2 \Rightarrow 1$).** Assume the Koopman operator commutes with the induced group action, i.e., $K_\pi \sigma = \sigma K_\pi$. For any observable $g$ and state $s$, this means $[K_\pi(\sigma g)](s) = [\sigma(K_\pi g)](s)$. We expand both sides using the definitions:

$$
\begin{aligned}
\text{LHS: } [K_\pi(\sigma g)](s) &= (\sigma g)(F_\pi(s)) = g(\sigma^{-1}(F_\pi(s))) \\
\text{RHS: } [\sigma(K_\pi g)](s) &= [K_\pi g](\sigma^{-1}(s)) = g(F_\pi(\sigma^{-1}(s)))
\end{aligned}
$$

Equating the expanded forms, we have:

$$g(\sigma^{-1}(F_\pi(s))) = g(F_\pi(\sigma^{-1}(s)))$$

This equality must hold for all observable functions $g$ in $\mathcal{K}(\mathcal{S})$. If the space of observables is sufficiently rich (e.g., separating points), this implies that the arguments of $g$ must be equal:

$$\sigma^{-1}(F_\pi(s)) = F_\pi(\sigma^{-1}(s))$$

Let $s' = \sigma^{-1}(s)$. This implies $s = \sigma \cdot s'$. Substituting $s'$ into the equation gives:

$$F_\pi(s') = \sigma^{-1}(F_\pi(\sigma \cdot s'))$$

Applying $\sigma$ to both sides yields:

$$\sigma \cdot F_\pi(s') = F_\pi(\sigma \cdot s')$$

Since this holds for any $s' \in \mathcal{S}$, the system is $\Sigma$-policy-equivariant. $\qquad\square$

## A.3 PROOF OF THEOREM 2 (GENERATING POLICY-CONSISTENT TRAJECTORIES)

**Theorem 5.** *Let $F_\pi$ be a $\Sigma$-policy-equivariant system with Koopman operator $K_\pi$ and let $\sigma \in \Sigma$. Let $\{z_t = E(s_t)\}$ be the latent representation of a trajectory under policy $\pi$, such that $z_{t+1} = K_\pi z_t$. Define a new trajectory $\{\hat{z}_t\}$ by $\hat{z}_t = \sigma \cdot z_t$. Then this new trajectory evolves according to the same Koopman dynamics: $\hat{z}_{t+1} = K_\pi \hat{z}_t$. Consequently, the decoded trajectory $\{\hat{s}_t = D(\hat{z}_t)\}$ is also a valid trajectory under the closed-loop dynamics of policy $\pi$.*

*Proof.* We are given an expert latent trajectory $\{z_t\}$ that follows the dynamics $z_{t+1} = K_\pi z_t$. The new trajectory is defined as $\{\hat{z}_t\}$ where $\hat{z}_t = \sigma \cdot z_t$. We aim to show that $\hat{z}_{t+1} = K_\pi \hat{z}_t$.

We start with the definition of the next state in the new trajectory, $\hat{z}_{t+1}$:

$$
\begin{aligned}
\hat{z}_{t+1} &= \sigma \cdot z_{t+1} &&\text{(by definition of the transformed trajectory)} \\
&= \sigma \cdot (K_\pi z_t) &&\text{(by the dynamics of the original trajectory)} \\
&= (\sigma K_\pi) z_t &&\text{(associativity of operator application)}
\end{aligned}
$$

Since the system is $\Sigma$-policy-equivariant, by Theorem 1, its Koopman operator commutes with the symmetry action: $\sigma K_\pi = K_\pi \sigma$. We substitute this into our derivation:

$$
\begin{aligned}
\hat{z}_{t+1} &= (K_\pi \sigma) z_t \\
&= K_\pi (\sigma \cdot z_t) &&\text{(associativity)} \\
&= K_\pi \hat{z}_t &&\text{(by definition of $\hat{z}_t$)}
\end{aligned}
$$

Thus, we have shown that $\hat{z}_{t+1} = K_\pi \hat{z}_t$. This means the transformed latent trajectory $\{\hat{z}_t\}$ evolves according to the same linear dynamics governed by $K_\pi$ as the original expert trajectory.

Consequently, if the decoder $D$ correctly maps latent states back to the state space, the sequence of states $\{\hat{s}_t = D(\hat{z}_t)\}$ represents a trajectory that is dynamically consistent with the learned model of the expert's closed-loop behavior. $\qquad\square$

## A.4 ROBUSTNESS ANALYSIS: BOUNDED ERROR FOR TRANSFORMED TRAJECTORIES

The theorems in the preceding sections assume an ideal, perfectly commuting Koopman operator and symmetry transformation. In practice, both our learned Koopman operator $\mathbf{K}$ and our transformation $\sigma$ are approximations, meaning their commutation is not perfect. This section provides a more general proof that accounts for this imperfection.

We define the **commutation error** operator, $\Delta$, as the commutator of $\sigma$ and $\mathbf{K}$:

$$\Delta \triangleq \sigma\mathbf{K} - \mathbf{K}\sigma$$

In an ideal scenario, $\Delta = 0$. In practice, $\Delta$ represents how much our learned symmetry fails to commute with our learned dynamics.

**Theorem 6** (Bounded Error for Transformed Trajectories). *Let $\{z_t^E\}$ be a latent expert trajectory with a sequence of one-step prediction errors $\{\epsilon_t\}$, where $\epsilon_t = z_{t+1}^E - \mathbf{K}z_t^E$. Let $\sigma$ be an approximately policy-equivariant transformation with commutation error $\Delta = \sigma\mathbf{K} - \mathbf{K}\sigma$. Then the one-step prediction error $\hat{\epsilon}_t$ for the transformed trajectory $\{\hat{z}_t = \sigma z_t^E\}$ is given by:*

$$\hat{\epsilon}_t = \sigma\epsilon_t + \Delta z_t^E. \tag{13}$$

*Consequently, its norm is bounded as:*

$$\|\hat{\epsilon}_t\| \leq \|\sigma\|\|\epsilon_t\| + \|\Delta\|\|z_t^E\|. \tag{14}$$

*Proof.* We begin with the definition of the prediction error for the transformed trajectory, $\hat{\epsilon}_t$:

$$\hat{\epsilon}_t \triangleq \hat{z}_{t+1} - \mathbf{K}\hat{z}_t$$

Substitute the definitions of the transformed trajectory, $\hat{z}_t = \sigma z_t^E$ and $\hat{z}_{t+1} = \sigma z_{t+1}^E$:

$$\hat{\epsilon}_t = \sigma z_{t+1}^E - \mathbf{K}(\sigma z_t^E)$$

Now, substitute the expression for the original trajectory's dynamics, $z_{t+1}^E = \mathbf{K}z_t^E + \epsilon_t$:

$$\hat{\epsilon}_t = \sigma(\mathbf{K}z_t^E + \epsilon_t) - \mathbf{K}\sigma z_t^E$$

By the linearity of the operator $\sigma$, we can distribute it over the sum:

$$\hat{\epsilon}_t = \sigma\mathbf{K}z_t^E + \sigma\epsilon_t - \mathbf{K}\sigma z_t^E$$

Rearranging the terms to group the components related to $\epsilon_t$ and $z_t^E$:

$$\hat{\epsilon}_t = \sigma\epsilon_t + (\sigma\mathbf{K} - \mathbf{K}\sigma)z_t^E$$

By our definition of the commutation error, $\Delta = \sigma\mathbf{K} - \mathbf{K}\sigma$, we arrive at the first result:

$$\hat{\epsilon}_t = \sigma\epsilon_t + \Delta z_t^E$$

This completes the proof of the first equation.

To derive the bound on the norm, we take a suitable matrix/vector norm (e.g., the L2 norm) of both sides and apply the triangle inequality ($\|a + b\| \leq \|a\| + \|b\|$):

$$\|\hat{\epsilon}_t\| = \|\sigma\epsilon_t + \Delta z_t^E\| \leq \|\sigma\epsilon_t\| + \|\Delta z_t^E\|$$

Finally, applying the property of induced matrix norms ($\|Ax\| \leq \|A\|\|x\|$) to each term gives the final bound:

$$\|\hat{\epsilon}_t\| \leq \|\sigma\|\|\epsilon_t\| + \|\Delta\|\|z_t^E\|$$

This completes the proof. $\square$

IMPLICATIONS OF THE BOUND

This result is highly significant. It shows that the error in a synthesized trajectory has two distinct sources:

1. **Propagated Model Error** ($\|\sigma\|\|\epsilon_t\|$)**:** The original model's inability to perfectly predict the expert dynamics ($\epsilon_t$) is carried over and scaled by the norm of the transformation.
2. **Symmetry Violation Error** ($\|\Delta\|\|z_t^E\|$)**:** A new error term arises directly from the failure of the learned symmetry to perfectly commute with the learned dynamics.

This provides a clear theoretical motivation for not only minimizing the standard reconstruction/prediction loss (which minimizes $\|\epsilon_t\|$), but also for explicitly regularizing the model to minimize the commutation error $\|\Delta\|$.

# B ROBUSTNESS IN FEW-SHOT SETTINGS

As demonstrated empirically in the Figure 4, our symmetry-based approach exhibits remarkable robustness to reductions in the size of the expert dataset, a regime where standard methods like Behavioral Cloning (BC) often fail. This appendix provides a qualitative analysis explaining this superior data efficiency.

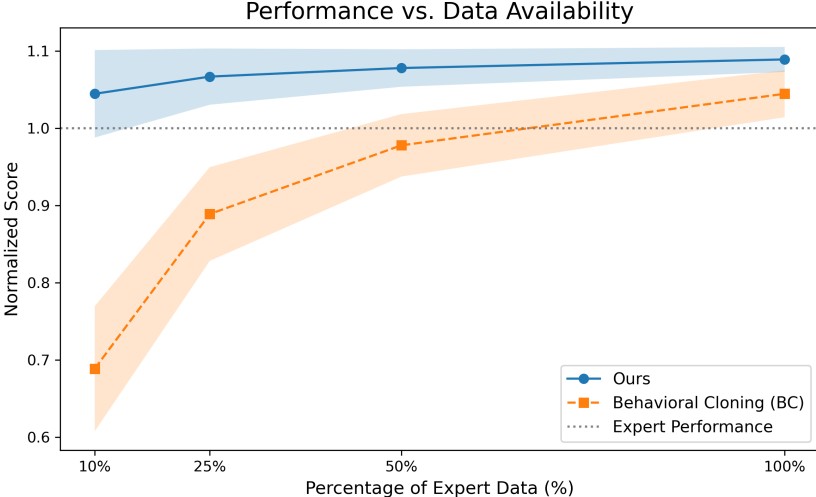

Figure 4: Few-Shot Performance Comparison on maze2D-umaze.

**The Brittleness of Standard Imitation Learning.** Standard Behavioral Cloning (BC) directly learns a state-action mapping from the provided expert data in a supervised manner. The performance of a BC policy is therefore critically dependent on the coverage of the state space by the training data. When the dataset is small, it represents a sparse and potentially biased sample of the true expert state distribution. This leads to two primary failure modes:

1. **Poor Generalization:** The learned policy performs well only in the immediate vicinity of the training states. It has no knowledge of how to act in the vast "in-between" regions of the state space.

2. **Covariate Shift:** A small initial error can push the agent into an out-of-distribution (OOD) state. Since the policy was not trained on data from this region, its subsequent actions are likely to be suboptimal or even random, leading to a cascade of compounding errors and rapid task failure.

**Data Augmentation as the Source of Robustness.** In contrast, our method is fundamentally a data augmentation technique that addresses the core issue of sparse data coverage. The key advantage stems from the learned symmetry transformation, $\sigma$. Even from a single expert trajectory, our model can generate a continuous family of new, diverse trajectories by applying the transformation pointwise: $\{\hat{s}_t\} = D_\psi(\sigma(E_\phi(s_t^E)))$.

Crucially, as proven in Appendix A, these augmented trajectories are guaranteed to be dynamically consistent. This process aims to densify the training data distribution, potentially bridging gaps in the state-space coverage that might otherwise hinder learning from sparse datasets. The policy is consequently trained on a much richer and more comprehensive representation of the system's behavior. This expanded dataset inoculates the policy against minor deviations, as states that would be OOD for BC are now likely to be in-distribution for our augmented dataset.

**Conclusion.** In summary, the performance of BC is directly coupled to the density and quantity of the provided expert samples. Our method decouples this dependency by leveraging the learned geometric and dynamic structure of the system (i.e., its symmetries) to synthetically generate a dense, dynamically valid dataset from a sparse source. This intrinsic data amplification is the primary reason for its superior performance and robustness in low-data regimes.

## C  ANALYSIS OF WEIGHT DISTRIBUTION AND ADAPTIVE SYMMETRY CONTROL

To investigate the interpretability and effectiveness of the learned weights in our framework, we analyze how these weights adaptively control the symmetry operator. The core objective is to ensure conservative behavior in data-sparse regions (where prediction error is high) while placing trust in augmentation within data-dense regions (where prediction is reliable).

We conducted an empirical validation on the Maze-2D dataset. Specifically, we sampled $N = 5,000$ state transitions and computed their corresponding weights derived from the Koopman prediction operator. We visualize the relationship between data density, prediction error, and weight magnitude using Principal Component Analysis (PCA).

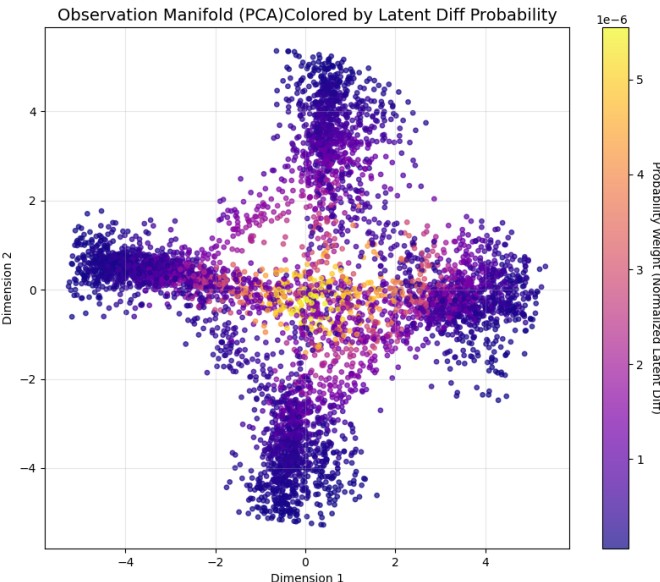

Figure 5: PCA visualization of the weight distribution on the Maze-2D dataset.

As illustrated in Figure 5, the results corroborate our hypothesis regarding the data-adaptive capability of the $\sigma$ training:

- **Data-Dense Regions:** Represented by darker colors in the visualization, these areas exhibit low prediction errors. Correspondingly, the model assigns small weights, implying that the generated augmentations are reliable and can be trusted.
- **Sparse Regions:** In the central areas of the maze where data is sparse, we observe high prediction errors. The model assigns large weights to these instances, necessitating a conservative update to prevent distribution shift.

Quantitatively, we measured the deviation of the learned weight distribution from a uniform distribution. The analysis yields a Kullback-Leibler (KL) divergence of $0.429$ and a Total Variation (TV) distance of $0.374$. These metrics confirm that the weights are non-uniform and effectively capture the underlying geometry and density of the dataset.

## D  EXPERIMENTAL DETAILS

This section provides a detailed breakdown of the model architectures, hyperparameters, and training procedures used in our experiments to ensure full reproducibility.

Our method, which we refer to as Koopman-Assisted Trajectory Synthesis for Offline Imitation Learning (KATS), consists of four sequential training stages: Koopman model pre-training, inverse dynamics model training, sigma model training, and finally behavior cloning with data augmentation.

**Model Architectures.** All neural networks in our framework are implemented as Multi-Layer Perceptrons (MLPs) with Tanh activation functions:

- **Koopman Model:** Consists of an encoder and decoder, both implemented as 3-layer MLPs with hidden dimensions of [512, 512]. The encoder maps from observation space to the Koopman latent space of dimension $N_z = 32$, while the decoder reconstructs observations from the latent representation. The linear Koopman operator $K \in \mathbb{R}^{32 \times 32}$ operates in the latent space.

- **Inverse Dynamics Model:** 3-layer MLP with hidden dimensions of [256, 256], taking concatenated current and next states as input and outputting predicted actions.

- **Sigma Model:** 3-layer MLP with hidden dimensions of [256, 256], mapping from latent space to latent space with the same dimensionality $N_z = 32$.

- **Policy Model:** 3-layer MLP with hidden dimensions of [256, 256], mapping from observation space to action space. For continuous action spaces, a tanh activation is applied to the final layer.

**Stage 1: Koopman Model Pre-training.** The Koopman model, which combines an encoder-decoder architecture with a linear Koopman operator $K$, is pre-trained to learn the underlying dynamics in latent space. The model is trained for up to 400 epochs with early stopping based on train-to-test loss ratio convergence (threshold: 1.0 for both reconstruction and Koopman losses). We use the ADAM optimizer with an initial learning rate of $3 \times 10^{-4}$, which is halved every 50 epochs after epoch 100. The batch size is set to 256, and 70% of the dataset is used for training with the remaining 30% for validation.

**Stage 2: Inverse Dynamics Model Training.** The inverse dynamics model is trained to predict actions given state transitions $(s_t, s_{t+1}) \rightarrow a_t$. This model is essential for generating corresponding actions for the augmented state trajectories. Training is conducted for 50 epochs using the ADAM optimizer with a learning rate of $1 \times 10^{-3}$ and the same train-test split as the forward model.

**Stage 3: Sigma Model Training.** The sigma model learns the transformation $\sigma : \mathcal{Z} \rightarrow \mathcal{Z}$ that enables controlled data augmentation in the latent space. The training objective is:

$$\mathcal{L}_\sigma = \mathbb{E}_{(z_t, z_{t+1})} \left[ w \cdot \| K\sigma(z_t) - \sigma(z_{t+1}) \|^2 \right]$$

where $w = \exp(\tau \| z_{t+1} - Kz_t \|^2)$ with $\tau = 1.5$. The model is trained for 30 epochs using the ADAM optimizer with a learning rate of $1 \times 10^{-4}$.

**Stage 4: Behavior Cloning with Data Augmentation.** The final stage trains the policy model using both original and augmented data. The data augmentation process works as follows:

1. Encode original states $(s_t, s_{t+1})$ into latent space: $z_t, z_{t+1}$

2. Apply sigma transformation: $\sigma(z_t), \sigma(z_{t+1})$

3. Decode back to state space: $\tilde{s}_t, \tilde{s}_{t+1}$

4. Use inverse dynamics model to predict corresponding actions: $\tilde{a}_t$

The policy is trained for 50 epochs using standard mean squared error loss for continuous actions or cross-entropy loss for discrete actions. We use the ADAM optimizer with a learning rate of $5 \times 10^{-4}$. When data augmentation is enabled (controlled by `use_data_augmentation` flag), the training alternates between original and augmented data with equal weighting.

**Hyperparameter Summary.**

- **Latent dimension:** $N_z = 32$
- **Batch size:** 256 across all training stages
- **Learning rates:** $3 \times 10^{-4}$ (Koopman model), $1 \times 10^{-3}$ (inverse model), $5 \times 10^{-4}$ (policy), $1 \times 10^{-4}$ (sigma model)
- **Training epochs:** Forward model (up to 400 with early stopping), Inverse model (50), Sigma model (30), Policy (50)
- **Sigma weighting parameter:** $\tau = 1.5$
- **Data augmentation weight:** shift_sigma $\in [0.1, 0.5]$, controls the weighted fusion between augmented and original data
- **Train-test split ratio:** 0.7/0.3

**Computational Resources.** All experiments were conducted using PyTorch with CUDA support. The models were trained on NVIDIA GeForce RTX 3090 GPUs with device selection controlled by the cuda_device parameter. Each complete training run (all four stages) typically requires 2-4 hours depending on the environment complexity and dataset size.

# E    USE OF LARGE LANGUAGE MODELS

A Large Language Model (LLM) was used to assist in the writing of this manuscript. Its role was exclusively for language polishing, including improving grammar, clarity, and style. The LLM had no role in the research ideation, methodological design, or data interpretation. The human authors are solely responsible for the intellectual content of this work.

