# OpenReview forum: "Koopman-Assisted Trajectory Synthesis: A Data Augmentation Framework for Offline Imitation Learning"
_ICLR.cc/2026/Conference — ICLR 2026 Poster_

### Official Review · Reviewer_qXWS · 2025-10-29

**Soundness:** 3
**Presentation:** 3
**Contribution:** 3
**Rating:** 8
**Confidence:** 3

**Summary:**

In this paper, a data augmentation framework, namely KATS that leverages Koopman theory to address the critical distribution shift problem in offline imitation learning is introduced. The presented framework can synthesize trajectories for the training data augmentation while avoiding the compounding errors of recursive rollouts and ensuring computational efficiency and scalability. Experimental results show the effectiveness of the proposed approach.

**Strengths:**

1. The paper is well-motivated and aims to address an important issue in the literature, which generates high-quality, dynamically consistent trajectory-level data while avoiding the compounding errors and ensuring computational efficiency.
2. Experimental results along with theoretical guarantees demonstrate the advantages of KATS, which yields substantial improvements in policy performance on some tasks.

**Weaknesses:**

1. The presented Koopman-Assisted Trajectory Synthesis (KATS) framework is based on the assumption that the symmetries of a closed-loop dynamical system, driven by a fixed expert policy, are directly reflected as commutation properties of its associated Koopman operator. While the assumption may hold in some cases, it is unclear whether such an argument can be satisfied in a general sense. Can the proposed framework be applied in any type of environment, or is its application limited to some special domains?
2. In the literature, many works have been proposed for the trajectory-level data augmentation, more recent baselines, especially for diffusion-based approaches, can be added and discussed for the comparison.
3. In the experiments, the most recent baseline is a rejected paper (TELS) while other baselines were mainly presented two or three years ago. Considering the rapid development of the related research area, it is necessary to adopt more recent baselines to verify the effectiveness of the proposed method.

**Questions:**

Please refer to the weakness points.

---

> ### Author Response · Authors · 2025-11-24
>
> We thank the Reviewer qXWS's valuable feedbacks. Here are some points we need to clarify:
>
> **W1:** The presented Koopman-Assisted Trajectory Synthesis (KATS) framework is based on the assumption that the symmetries of a closed-loop dynamical system, driven by a fixed expert policy, are directly reflected as commutation properties of its associated Koopman operator. While the assumption may hold in some cases, it is unclear whether such an argument can be satisfied in a general sense. Can the proposed framework be applied in any type of environment, or is its application limited to some special domains?
>
> **Response:**
>
> We thank the reviewer for this insightful comment. We would like to clarify that our framework is rigorously grounded in the symmetry properties of the closed-loop system.
>
> To provide a precise derivation, we formally distinguish between the symmetry transformations applied to the state and action spaces:
> *   Let $T_s: \mathcal{S} \rightarrow \mathcal{S}$ denote the transformation on the state space.
> *   Let $T_a: \mathcal{A} \rightarrow \mathcal{A}$ denote the corresponding transformation on the action space. $T_a$ represents the necessary transformation of control inputs to maintain consistency with the underlying symmetry group (e.g., if $T_s$ mirrors the robot's pose, $T_a$ mirrors the applied forces).
>
> The core commutation assumption holds if the closed-loop system is **equivariant**. Defining the environment dynamics as $s' = f(s, a)$ and the policy as $a = \pi(s)$, the closed-loop transition $F(s) = f(s, \pi(s))$ satisfies the equivariance condition $F(T_s(s)) = T_s(F(s))$ provided that two conditions are met:
>
> 1.  **Dynamics Equivariance:** The physical laws are invariant under the joint transformation, i.e., $f(T_s(s), T_a(a)) = T_s(f(s, a))$.
> 2.  **Policy Equivariance:** The policy acts consistently with the symmetry, i.e., $\pi(T_s(s)) = T_a(\pi(s))$.
>
> **Derivation:**
> Substituting these conditions into the closed-loop dynamics demonstrates the commutation:
>
> $$
> \begin{aligned}
> F(T_s(s)) &= f(T_s(s), \pi(T_s(s))) & \quad \text{(Definition of closed-loop)} \\\\
> &= f(T_s(s), T_a(\pi(s))) & \quad \text{(By Cond. 2: Policy Equivariance)} \\\\
> &= T_s(f(s, \pi(s))) & \quad \text{(By Cond. 1: Dynamics Equivariance)} \\\\
> &= T_s(F(s)) & \quad \text{(Q.E.D.)}
> \end{aligned}
> $$
>
> This derivation confirms that in physical control domains (e.g., legged locomotion, manipulation), where physics enforces dynamics equivariance and optimal agents naturally exhibit policy equivariance, the closed-loop system preserves the symmetry structure, thereby justifying the applicability of our method.
>
> **W2,W3:** Comparison with more recent baselines, particularly diffusion-based trajectory augmentation approaches, is necessary to verify the effectiveness of the proposed method.
>
> **Response:**
>
> We thank the reviewer for the valuable suggestion. We agree that comparing against diffusion-based methods is essential for a rigorous evaluation. Accordingly, we have added Diffuser as a new baseline in our experiments. This allows us to benchmark KATS against the latest state-of-the-art in trajectory generation. The updated results show that KATS maintains superior performance and efficiency, effectively demonstrating the advantages of our symmetry-based approach compared to diffusion-based generative models.
>
> The performance comparison on the 10% data subset (using the Table 2 setting) is shown below:
>
> | Task | Diffuser | **KATS+BC (Ours)** |
> | :--- | :---: | :---: |
> | Hopper-e | 74.5$\pm$3.8 | **97.1$\pm$10.4** |
> | Hopper-me | 70.1$\pm$6.5 | **87.2$\pm$16.5** |
> | Halfcheetah-e | 68.9$\pm$4.1 | **102.7$\pm$8.1** |
> | Halfcheetah-me | 63.2$\pm$8.8 | **75.2$\pm$10.4** |
> | Walker2d-e | 83.3$\pm$6.7 | **109.4$\pm$11.2** |
> | Walker2d-me | 74.2$\pm$9.0 | **92.5$\pm$8.1** |
> | Antmaze-u-d | 59.1$\pm$11.7 | **67.5$\pm$10.3** |
> | Antmaze-u | 62.1$\pm$9.4 | **83.1$\pm$7.2** |
> | Antmaze-m-d | 51.4$\pm$8.7 | **60.3$\pm$13.8** |
> | Antmaze-m-p | 60.9$\pm$12.6 | **71.3$\pm$15.2** |
> | Antmaze-l-d | 38.5$\pm$10.9 | **42.8$\pm$11.9** |
> | Antmaze-l-p | 47.6$\pm$8.3 | **51.4$\pm$13.1** |
>
> If you have any further questions, we are happy to continue the discussion:)

---

### Official Review · Reviewer_AWKq · 2025-10-31

**Soundness:** 3
**Presentation:** 2
**Contribution:** 3
**Rating:** 8
**Confidence:** 3

**Summary:**

This paper proposes a theoretically principled method to generate augmented, synthetic expert demonstrations for offline imitation learning. With said augmented data, one can perform standard behavioral cloning on the union of the given expert dataset and the augmented dataset. The paper grounds their technique for augmenting trajectories in Koopman theory, done in the latent space of a learned autoencoder over the expert data. In particular, they note that if the learned state latents satisfy certain linear properties (e.g. the transition function under the expert data is linear -- said linear transformation is the Koopman operator), then compounding errors are bounded, leading to useful trajectory generation.

Experiments on MuJoCo tasks, both in IL and RL, validate the hypothesis shown, including strong results compared to prior offline IL baselines that either had to rely on suboptimal offline data (e.g. MILO) and other methods employing data augmentation (e.g. KFC+BC). Furthermore, they also try their method in the offline RL setting, comparing to standard baselines and recent Koopman-focused baselines such as KFC++ and showing strong performance.

**Strengths:**

I really like this area of work, as Koopman theory is well grounded and has been used for quite a while in model-based dynamical system control (although I am not super familiar with the literature). There are many advantages in learning a latent space under which the transition dynamics are linear, which include not even having to do RL directly and employing more stable control theory-focused algorithms.

The theory seems fine to me, and seems to borrow a lot from the KFC paper (Weissenbacher et al. 2022), leading me to believe that it is sound. The experimental results are also strong, showing strong performance improvement even over other Koopman-based RL and IL algorithms.

**Weaknesses:**

There are instances where the paper could be written a bit better (e.g. put your citations in parentheses!). I think there are also potential weaknesses to the method empirically, which include (and correct me if I'm wrong) the following:

- In the RL setting, if the reward function is a function of both state and action, then Q learning may be biased. I think that the reward is only the function of the state in MuJoCo and DMC control domains, which means the method is fine there, but in cases where it is not, then I figure due to said bias, learning is difficult even when the latent system is well-learned.

- Generally, these methods seem to work in small-scale tasks such as OpenAI Gym locomotion, while not having been tested on larger-scale domains such as DMC from pixels or larger control tasks. A potentially great use for this method could be to learn such a linear latent system on real robotic datasets, making learning controllers much faster.

These are not "make or break" weaknesses, more so that this seems not to have been tested. In general, for the focus of IL, there are less problems, as for instance removing action conditioning is fine as the Markovian expert policy is embedded into the latent encoding, which is enough for data augmentation.

**Questions:**

No fundamental questions from me, but I am curious to know if any large-scale experiments were done for this paper, including with either high-dimensional states or with image observations.

---

> ### Author Response · Authors · 2025-11-24
>
> We thank the Reviewer AWKq's valuable feedbacks. Here are some points we need to clarify:
>
> **W1:** In the RL setting, if the reward function is a function of both state and action, then Q learning may be biased. I think that the reward is only the function of the state in MuJoCo and DMC control domains, which means the method is fine there, but in cases where it is not, then I figure due to said bias, learning is difficult even when the latent system is well-learned.
>
> **Response:**
> We thank the reviewer for highlighting the theoretical implications of action-dependent rewards, which is indeed a crucial consideration in broader offline RL.
>
> We would like to clarify that our method is primarily designed as a data augmentation technique for Imitation Learning (IL). Our core framework, KATS, generates kinematically consistent trajectories $(s', a')$ to train Behavioral Cloning (BC) agents. Since this training process is entirely reward-free, the complexity of the reward function—whether $r(s)$ or $r(s, a)$—does not introduce bias into our framework. Our experiments focus on this IL setting to demonstrate the efficacy of the augmentation strategy itself.
>
> However, we agree that extending symmetry-based augmentation to handle action-dependent rewards is a valuable direction for future research (e.g., by learning a reward model $R(s, a)$ that respects transformation properties). We will discuss this scope and future direction in the final manuscript.
>
> **W2:** Generally, these methods seem to work in small-scale tasks such as OpenAI Gym locomotion, while not having been tested on larger-scale domains such as DMC from pixels or larger control tasks. A potentially great use for this method could be to learn such a linear latent system on real robotic datasets, making learning controllers much faster.
>
> **Response:**
> We thank the reviewer for the valuable feedback on scalability. Regarding our experimental setting, we prioritized state-based MuJoCo tasks as they are the standard benchmarks for offline policy learning, facilitating direct comparison with prior art.
>
> However, the reviewer’s points regarding broader applications align perfectly with our future roadmap:
>
> - **High-Dimensional Inputs:** Applying KATS to pixel-based inputs is a logical progression. This would involve integrating visual representation learning to extract state features, raising interesting questions about jointly learning representations and symmetries.
> - **Real Robotic Datasets:** We agree that KATS is uniquely suited for real-world robotics, where data is scarce and expensive. Our trajectory augmentation can significantly boost sample efficiency, and the use of linear latent dynamics offers strong potential for accelerating hardware transfer.
>
> We will add a discussion on these limitations and future directions in the final version of the paper.
>
> If you have any further questions, we are happy to continue the discussion:)

---

### Official Review · Reviewer_f7Wk · 2025-10-31

**Soundness:** 3
**Presentation:** 3
**Contribution:** 4
**Rating:** 6
**Confidence:** 4

**Summary:**

The paper tackles covariate shift in offline imitation learning, where agents are limited to fixed datasets with potentially low diversity. Prior Koopman-based methods operate at the single-step level, causing dynamic inconsistency and high cost. Koopman-Assisted Trajectory Synthesis (KATS) is introduced as a trajectory-level data augmentation method that generates novel yet dynamically consistent expert-like trajectories in a learned state-equivariant linear Koopman latent space. In addition, KATS is adaptive in that it prioritizes data synthesis where the model is uncertain. Theoretical results guarantee that symmetries commuting with the learned Koopman operator yield trajectories consistent with expert policy dynamics. In practice, KATS augments data and then applies simple BC, outperforming more complex offline IL and RL baselines. KATS serves as a plug-and-play augmentation module that enhances existing algorithms through high-fidelity, behaviorally consistent data generation.

**Strengths:**

* Th. 2 is very elegant and clear. Slight suggestion: preamble the section with the gist/a teaser of what the theorem will show.
* Sec. 5.2: ingenious yet simple.
* KATS demonstrates that augmenting data and applying behavioral cloning can be a more effective and reliable strategy for imitation, even in data-scarce regimes where traditional apprenticeship learning methods, despite allowing interaction, tend to be fragile and overcomplicated.

**Weaknesses:**

* Th. 3 should be followed by an "in-words" interpretation and description of its consequences, along with a hyperlinked reference to the “Implication of the Bound” section presented in the Appendix.
* KATS is introduced from the machinery of KFC, but the link is somewhat lost until it comes back in 5.1 line 287, where it is clear.
* The way the authors present sigma in Fig 2 (“Sigma 0.2, Sigma 0.3” etc.) or as “a symmetry basis” suggests a structured family of symmetry operators or scalars, but in the actual text and appendix, they never describe how that basis or scaling is obtained. The figure’s depiction of multiple sigma’s or scaled sigma’s is not grounded in the described theory or implementation. The authors must explicitly indicate how they operationalize the learned sigma network to obtain their basis.
* The authors write: "This dramatic leap provides strong validation for our core contribution: the action-independent formulation.". It is definitely noteworthy and interesting to observe that the action-independent closed-loop formulation combined with an IDM can yield such results. Stating that the generated trajectories are "by construction, more behaviorally consistent than those from action-conditioned models like KFC" might be a bit of a stretch however, but an experimental design could be devised to showcase that further.
* In the appendix L795-796, the authors write "This process effectively densifies the training data, filling the gaps in the state-space coverage that a sparse dataset would otherwise leave open.". That is “potentially” what KATS enables, but the coverage is not proven or showcased as such.
* The paper could use an additional round of polish to remove the inconsistencies in notations.

**Questions:**

* Can the authors make it clear what they mean by scalability/scalable? From the phrasing in the paper, it seems that the scalability claimed by the authors is on the complexity of the task (dimension, degrees of freedom). However, since this is an approach for the low-data regime (limited offline dataset), it might be on the how the developed data augmentations impact performance, etc.
* Koopman theory imposes a strong inductive bias, going against the bitter lesson. Would the authors defend that modeling the temporal relationship between one latent and its successor simply with a linear operator is enough to model system that are more complex than the ones tackled empirically in the paper? In other words, would adding depth to the encoder always be enough to go in a sufficiently “higher-dimensional space” where the dynamics can reasonably be assumed linear?
* It is unclear in the text why KFC working at the single-step level is costly in compute (L193). Is it clearer costly because the action-equivariant assumption in KFC requires one linear operator per action dimension (Eq. 2), making it costly to scale with action dimensionality?
* Section 5.1: modeling the closed-loop system dynamics is a design choice rather than an innovation, is it not? Do the other claim it is an innovation because it is unusual?
* Is there a particular reason why the authors write "find symmetry basis" at line 2 of the algorithm, and not "learn the symmetries" ("sigma model training" only appears in the appendix, L812-813)? By basis or symmetries, do the author mean that the set verifies the properties of basis, then they then use to craft other symmetries as linear combinations of the elementary symmetries of the basis?
* What would be informative, for the data augmentation methods, is to get the final size of the dataset used to train the policy with BC (compared to the initial size), along with statistics that could give an idea as to how it expands the initial one in terms of diversity.
* Could the authors include a few words about the baselines, to get the gist of their approach, which would put KATS in perspective; e.g., how does TELS' data augmentation approach differ from KATS?
* The authors write in the appendix that the "[policy] training alternates between original and augmented data with equal weighting". What is the only sampling strategy that the authors tried? Could the authors give the respective dataset sizes for the reader to be able to gauge how likely to overfit the policy is?
* Have the authors experimented with learning the IDM from the mapping of the expert states into the learned latent space, i.e. from the latents z instead of the states s?
* What does the weight distribution (in the sigma loss) look like? In other words, how far from uniform is the "adaptive" scheme the authors designed?
* Providing an ablation study comparing KATS with and without any symmetry training and usage would be insightful.

Style, typos, suggestions:
* It would be useful to add, in the algorithm, links to the equation according to which the various networks are optimized.
* [minor] It might be good to mention that that Koopman machinery is used in latent space earlier in the introduction than at the very end. The paper might also benefit from putting KATS in the context of model learning in latent spaces, which the typical RL literature reader might be more familiar with.
* [minor] L123-124: the comma after "shift" should be removed.
* [minor] L155-156: z_t and z_{t+1} correspond to a transition (s_t, s_{t+1}) or to a pair of states but not a pair of transitions.
* [minor] L187-188: why use "aug" when the figure 1 (a) uses primes to designates the augmentations?
* [minor] The last sentence of Def. 2 should be emphasized.
* [minor] L263: properly format the emdash, or use a colon.
* [minor] End of page 9: "Limitaitions" -> Limitations
* [minor] L810-811: "Synthesisi" (extra "i"), "(KATS)" (missing space prefix).
* [minor] L289-290: "Any symmetry transformation that commutes with K is therefore guaranteed to produce trajectories that adhere to this policy." I find this to not be clear.

---

> ### Author Response · Authors · 2025-11-24
>
> We thank the Reviewer f7Wk's valuable feedbacks. Here are some points we need to clarify:
>
> **Q3,Q4:** Why is KFC considered costly, and why is modeling closed-loop dynamics an innovation?
>
> **Response:**
> We thank the reviewer for this very helpful question.
> Regarding the computational cost of KFC, the primary bottleneck arises from its action-equivariant assumption, where symmetry (or Koopman) operators are tightly coupled to specific actions. Concretely, this implies that distinct actions correspond to distinct linear operators, causing the number of effective operators—as well as the associated parameters and FLOPs per forward step—to grow at least linearly with the action dimension. Furthermore, because these symmetry operators are highly action-dependent, any variation in action generally necessitates the application of a different operator, thereby increasing the computational overhead.
>
> We acknowledge that modeling closed-loop dynamics is a design choice; however, the innovation lies in its application to **trajectory-level symmetry learning**, which distinguishes our method from prior step-wise augmentation approaches. Our primary motivation for adopting this sequence-to-sequence formulation is to overcome the limitations of KFC. Beyond improved computational efficiency, this design enables a more accurate approximation of the stationary distribution $d_\pi(s)$, which is crucial for capturing the long-term patterns required to mitigate covariate shift.
>
> **Q2:** (Koopman Theory vs. The Bitter Lesson) Is the linear operator assumption sufficient for complex systems?
>
> **Response:**
> We thank the reviewer for this profound question. We do not view our approach as contradicting the Bitter Lesson, but rather as a form of *Structured Deep Learning* that leverages massive computation to discover physically meaningful latent spaces. Our position rests on two key points:
>
> **1. the Bitter Lesson.**
> The Bitter Lesson prioritizes general-purpose computation (search and learning) over hand-crafted heuristics. We adhere to this principle by avoiding hand-crafted features. Instead, we employ a high-capacity deep encoder ($\psi$) to *search* for a representation governed by linear dynamics. While linearity serves as the architectural objective, we rely entirely on the raw computational power of deep learning to discover the complex transformation that satisfies this constraint.
>
> **2. the linear operator assumption.**
> A linear operator is sufficient for complex systems because Koopman-based learning shifts complexity from the *dynamics* to the *observation map* (the encoder). Theoretically, Koopman operator theory guarantees the existence of a linearizing space; practically, the Universal Approximation Theorem implies that a deep encoder can approximate this mapping to capture the system's dominant spectral properties. Thus, we are not modeling complex systems with simple tools; rather, we use a complex non-linear encoder to simplify the representation of the dynamics.
>
> **W3,Q5:** The reviewer notes that the terms "symmetry basis" and "Sigma 0.2" in Fig. 2 are not clearly defined in the text. How is the symmetry operationalized?
>
> **Response:**
> We appreciate the opportunity to clarify.
>
> In our implementation, we do not learn a single symmetry operator, but rather an **ensemble of symmetry operators** ${\sigma^{(1)}, \sigma^{(2)}, \dots, \sigma^{(M)}}$, where each $\sigma^{(m)}$ is a neural network mapping latent states to latent states. All $\sigma^{(m)}$ are trained with the same commutation loss $\mathcal{L}{\sigma}$ to approximately satisfy $\sigma^{(m)} K_\pi \approx K_\pi \sigma^{(m)}$. While we informally referred to this set as a 'symmetry basis' in the original text, we will clarify in the revision that this refers to a **learned ensemble of operators** rather than a strict linear-algebraic basis.
>
>
>
> The labels '$\sigma0.2$' and '$\sigma0.3$' in Figure 2 do not denote different operators. Instead, they represent different **interpolation strengths** ($\alpha$) between the original state and its symmetry-transformed version. Specifically, given a latent state $z_t$ and a symmetry operator $\sigma^{(m)}$, we generate an augmented state using:
> $\tilde{z}_t^{(\alpha, m)} = (1 - \alpha)\, z_t + \alpha\, \sigma^{(m)}(z_t), \quad \alpha \in [0, 1].$
> Thus, '$\sigma0.2$' corresponds to $\alpha = 0.2$ (shifting 0.2 toward the symmetry-transformed state), '$\sigma0.3$' corresponds to $\alpha = 0.3$, and so on.
>
> We will update the main text and the caption of Figure 2 to explicitly define the ensemble structure and the interpolation formula, ensuring the terminology is operationally precise.
>
> **W5:** Unverified Coverage
>
> **Response:**
> We acknowledge that the "densifying" effect was not explicitly verified. We will revise the text to state that the approach "potentially enhances effective state-space coverage," clarifying that this is a theoretical motivation rather than an empirical result.

---

> ### Author Response · Authors · 2025-11-24
>
> **Q11:** An ablation comparing KATS with/without symmetry training is needed.
>
> **Response:**
> We thank the reviewer for this valuable suggestion. In accordance with your suggestion, we performed additional ablation studies to validate the core hypotheses of the KATS method.
> *   **KATS (Full Method):** Integrates the learned $\sigma_\theta$-network for adaptive symmetry weights and the Inverse Dynamics Model (IDM) for trajectory-consistent action prediction.
> *   **KATS-$\sigma_A$:** Replaces the learnable $\sigma_\theta$ with a fixed analytical matrix $\sigma_A$ (derived via the Sylvester equation). This tests the value of learning flexible symmetries over rigid analytical ones.
> *   **KATS w/o IDM:** Removes the IDM, pairing augmented states with original actions. This isolates the necessity of the IDM in resolving state-action mismatches.
> *   **KFC:** Replaces our state-centric encoder $K_\pi$ with the action-coupled encoder from the KFC framework, validating our decoupled design choice.
>
> **Results:** As shown in the following table, the full KATS model significantly outperforms all variants. The performance drop in 'w/o IDM' confirms that action correction is critical, while the superiority over 'KATS-$\sigma_A$' highlights the necessity of the data-adaptive $\sigma_\theta$-network. We have included this analysis in the Appendix.
>
> **Table: Ablation study on the `maze2d` and `antmaze` environment. We report the normalized score (mean $\pm$ std over 5 seeds).**
>
> | Domain | Task Name | BC | KFC+BC | KATS w/o IDM | KATS-$\sigma_A$ | KATS |
> | :--- | :--- | :---: | :---: | :---: | :---: | :---: |
> | **AntMaze** | antmaze-umaze | $74.0 \pm 1.2$ | $79.1 \pm 3.4$ | $78.4 \pm 4.8$ | $87.3 \pm 2.6$ | **96.9 ± 0.8** |
> | | antmaze-umaze-diverse | $64.0 \pm 2.0$ | $66.2 \pm 1.5$ | $72.3 \pm 3.6$ | $82.1 \pm 3.3$ | **90.1 ± 0.7** |
> | | antmaze-medium-play | $68.2 \pm 3.1$ | $72.3 \pm 2.5$ | $69.5 \pm 2.8$ | $78.4 \pm 2.1$ | **82.7 ± 1.4** |
> | | antmaze-medium-diverse | $53.7 \pm 4.5$ | $57.1 \pm 2.2$ | $58.2 \pm 6.8$ | $65.2 \pm 4.5$ | **67.3 ± 2.5** |
> | | antmaze-large-play | $35.8 \pm 2.2$ | $42.8 \pm 1.5$ | $40.2 \pm 4.3$ | $51.7 \pm 4.0$ | **59.3 ± 1.5** |
> | | antmaze-large-diverse | $24.9 \pm 1.8$ | $28.1 \pm 1.7$ | $27.5 \pm 3.1$ | $36.7 \pm 2.6$ | **44.2 ± 2.7** |
> | **Maze2d** | maze2d-umaze | $72.1 \pm 6.0$ | $89.2 \pm 4.8$ | $86.6 \pm 8.1$ | $100.2 \pm 6.7$ | **113.2 ± 5.6** |
> | | maze2d-medium | $42.3 \pm 9.9$ | $63.7 \pm 8.5$ | $66.3 \pm 12.1$ | $90.8 \pm 11.5$ | **108.7 ± 4.9** |
> | | maze2d-large | $16.3 \pm 7.3$ | $55.9 \pm 9.1$ | $41.5 \pm 15.8$ | $89.2 \pm 12.2$ | **100.1 ± 7.2** |
>
>
> **Q9:** Why IDM on states?
>
> **Response:**
> We appreciate this insightful question regarding the input space for the IDM. In our preliminary experiments, we chose to learn the IDM in the raw state space primarily for the following two reasons:
>
> 1. **Dimensionality:** Koopman theory generally necessitates lifting the state space to a higher-dimensional manifold (where $\dim(z) > \dim(s)$) to ensure the global linearity of the dynamics. Training the IDM on this high-dimensional latent space significantly increases the complexity and parameter count of the network.
>
> 2. **Signal Fidelity and Noise:** The latent variable $z$ is an approximation produced by the learned encoder, optimized primarily for satisfying linear dynamics rather than preserving all control-relevant information. Consequently, $z$ may contain artifacts or accumulated errors from the encoder. By contrast, the raw state $s$ represents the ground truth observation. Using $s$ directly decouples the IDM from the encoder's reconstruction error, providing a cleaner and more robust signal for action inference.
>
> Per your advice, in the revised version, we plan to add new experimental results to compare the performance of the inverse dynamics model in the raw state space against that in the learned latent space.

---

> ### Author Response · Authors · 2025-11-24
>
> **Q10:** What does the weight distribution (in the sigma loss) look like? In other words, how far from uniform is the "adaptive" scheme the authors designed?
>
> **Response:**
> Thank you for the question. We use weights to adaptively control the symmetry operator: ensuring conservative behavior in sparse regions and trusting augmentation in dense regions.
>
> Per your advice, we validated this on the Maze-2D dataset by sampling 5,000 instances and computing their weights via the Koopman prediction operator. The PCA visualization confirms our hypothesis: data-dense regions (darker) show low errors and small weights, implying reliable augmentation, while sparse central regions show high errors and large weights, necessitating a conservative approach. This demonstrates the data-adaptive capability of the $\sigma$ training.The visualization figure can be found at the following link: https://anonymous.4open.science/r/KATS-A3EF/.
>
> Quantitatively, the weight distribution deviates from uniformity with a KL divergence of $0.429$ and a TV distance of $0.374$.
>
>
> **Q6,Q8:** Please provide final dataset sizes and sampling details.
>
> **Response:**
> In terms of dataset scale, for standard D4RL benchmarks (which typically contain **1 M transitions**), our method expands the original dataset by a factor of **8x**, resulting in a final training set of approximately 9 M transitions. Regarding the usage of this data, we do not employ a specific balancing heuristic; instead, the augmented data is combined with the original data and trained with **equal weight** in the objective function. This uniform treatment maximizes the diversity gain, allowing the policy to fully exploit the symmetry-aware expansions across the state space.
>
> **Q1:** Can the authors make it clear what they mean by scalability/scalable?
>
> **Response:**
> We thank the reviewer for requesting this clarification. The reviewer is correct that our primary claim of "scalability" refers to the method's ability to handle the complexity of the task, specifically regarding **high-dimensional action spaces** (high degrees of freedom).
>
> Unlike prior Koopman-based methods (e.g., KFC) which couple state and action spaces—leading to parameter explosion as dimensions grow—our approach employs an **action decoupling** strategy. This design ensures that the complexity of the Koopman operator remains manageable regardless of the action size. Consequently, it significantly reduces both operator training costs and memory requirements, enabling the method to scale efficiently to complex control tasks with high action dimensionality.
>
> **Q7:** Could the authors include a few words about the baselines?
>
> **Response:**
> We thank the reviewer for the suggestion. To better contextualize our comparisons, we have grouped the baselines according to the experimental setting:
>
> *   **Table 1 (Reward-Free Imitation Learning):** In this setting, we provide an apples-to-apples comparison of augmentation strategies. We compare KATS against **MILO** (generative adversarial IL) and **SRA**, a method that employs inverse dynamics to project out-of-distribution data back to the in-distribution manifold.
>
> *   **Table 2 (Reward-Based Offline RL):** Here, we aim to demonstrate that our trajectory-level augmentation provides sufficient signal to match methods that explicitly use rewards. We compare against classic RL algorithms (**CQL**, **IQL**) and dynamics-aware methods, specifically **POR** (a state-value guided approach) and **TELS** (which utilizes time-reversible dynamics models for regularization).
>
> **W1,W2,W6,Style,typos,suggestions:** Issues on Formatting, Notation, and Presentation.
>
> **Response:**
> We sincerely thank the reviewer for their careful reading and for pointing out these details. We apologize for the oversight. We will conduct a thorough proofreading of the paper to unify the notation, correct the formatting inconsistencies, and polish the writing. All these changes will be incorporated into the revised manuscript.
>
> **W1:** The action-independent formulation is "by construction" more behaviorally consistent than action-conditioned models is an overstatement.
>
> **Response:**
> We thank the reviewer for this valid point and agree that the phrase "by construction" is too strong. Our intention was to emphasize that decoupling state generation from action inference prevents the compounding errors common in action-conditioned models (like KFC), where action deviations can drift states out-of-distribution. In the revised manuscript, we will clarify that our formulation "mitigates compounding errors associated with joint state-action generation," rather than guaranteeing consistency. We believe the improved downstream performance empirically supports this stability.
>
> If you have any further questions, we are happy to continue the discussion:)

---

### Official Review · Reviewer_cuKL · 2025-11-01

**Soundness:** 3
**Presentation:** 2
**Contribution:** 2
**Rating:** 4
**Confidence:** 3

**Summary:**

This paper proposes a method based on Koopman Theory for generating trajectories from offline data.

**Strengths:**

The approach appears reasonably sound.

**Weaknesses:**

1. (I have not read KFC) The authors claim that this work differs from KFC, as KFC only generates single-step data, while KATS generates trajectories. However, judging from Equations 7, 8, 9, and 10, KATS still appears to generate states.

2. Although the experimental results presented by the authors show that KATS performs well, the experiments seem insufficient. For example, there is no ablation study.

3. The baselines compared in Table 1 and Table 2 are inconsistent:
(1) Table 1 compares KATS+BC and KFC+BC, while Table 2 compares KATS+BC and KFC+CQL. Since the base algorithms of KATS+BC and KFC+CQL are different, the comparison lacks fairness.
(2) In Table 1, the data augmentation methods compared are SRA, MOLI, and KFC+BC, while in Table 2, the compared methods are TELS, DOGE, POR, and KFC+CQL.

**Questions:**

1. What are the differences and connections between Equations 7–8 and 9–10? In implementation, are they used together or only 9–10?

2. Why does Table 1 compare KATS+BC with KFC+BC, while Table 2 compares KATS+BC with KFC+CQL?

3. Why are the data augmentation methods in Table 1 compared with SRA, MOLI, and KFC+BC, while in Table 2, they are compared with TELS, DOGE, POR, and KFC+CQL?

---

> ### Author Response · Authors · 2025-11-24
>
> We thank the Reviewer cuKL's valuable feedbacks. Here are some points we need to clarify:
>
> **W1:** (I have not read KFC) The authors claim that this work differs from KFC, as KFC only generates single-step data, while KATS generates trajectories. However, judging from Equations 7, 8, 9, and 10, KATS still appears to generate states.
>
> **Response:**
> We thank the reviewer for the question. While both methods train on transitions, KATS enables **trajectory-level synthesis** through its closed-loop formulation, whereas KFC is restricted to **single-step augmentation**.
>
> KFC relies on an action-dependent operator $\mathbf{K}(a_t)$. Since the augmented policy is unknown during generation, KFC cannot synthesize future actions and is limited to locally perturbing existing $(s, a, s')$ tuples. In contrast, KATS models closed-loop dynamics ($\mathbf{K}$), allowing us to apply the learned symmetry operator $\sigma_\theta$ to an entire expert trajectory $\{z_t\}$ at once. The commutation property ($\sigma \mathbf{K} = \mathbf{K} \sigma$) guarantees that the globally transformed sequence $\{\sigma_\theta(z_t)\}$ remains dynamically consistent throughout the horizon. We will revise the manuscript to explicitly distinguish between the local training objective and this global generation mechanism.
>
> **Q1:** What are the differences and connections between Equations 7–8 and 9–10? In implementation, are they used together or only 9–10?
>
> **Response:**
> We appreciate the opportunity to clarify that our training pipeline is **sequential**, not simultaneous. Equations (7–8) and (9–10) correspond to distinct phases of optimization:
>
> *   **Dynamics Learning (Eqs. 7–8):** First, we train the Autoencoder and Operator $\mathbf{K}$ to establish the system's latent dynamics.
> *   **Symmetry Discovery (Eqs. 9–10):** Next, we **freeze** the dynamics model (Encoder, Decoder, and $\mathbf{K}$) and optimize only the symmetry operator $\sigma$. The goal is to find transformations that commute with the fixed operator $\mathbf{K}$.
>
> This hierarchical approach ensures that symmetries are discovered with respect to a stable dynamical system. We will revise Sections 5.1 and 5.2 to explicitly describe this sequential optimization process and prevent any confusion regarding the loss landscape.
>
> **W2:** Although the experimental results presented by the authors show that KATS performs well, the experiments seem insufficient. For example, there is no ablation study.
>
> **Response:**
> We thank the reviewer for this suggestion. We have added a detailed ablation study on the `maze2d` and `antmaze` environments to validate the contribution of each component:
>
> *   **KATS (Full Method):** Integrates the learned $\sigma_\theta$-network for adaptive symmetry weights and the Inverse Dynamics Model (IDM) for trajectory-consistent action prediction.
> *   **KATS-$\sigma_A$:** Replaces the learnable $\sigma_\theta$ with a fixed analytical matrix $\sigma_A$ (derived via the Sylvester equation). This tests the value of learning flexible symmetries over rigid analytical ones.
> *   **KATS w/o IDM:** Removes the IDM, pairing augmented states with original actions. This isolates the necessity of the IDM in resolving state-action mismatches.
> *   **KFC:** Replaces our state-centric encoder $K_\pi$ with the action-coupled encoder from the KFC framework, validating our decoupled design choice.
>
> **Results:** As shown in the following table, the full KATS model significantly outperforms all variants. The performance drop in 'w/o IDM' confirms that action correction is critical, while the superiority over 'KATS-$\sigma_A$' highlights the necessity of the data-adaptive $\sigma_\theta$-network. We have included this analysis in the Appendix.
>
> | Domain | Task Name | BC | KFC+BC | KATS w/o IDM | KATS-$\sigma_A$ | KATS |
> | :--- | :--- | :---: | :---: | :---: | :---: | :---: |
> | **AntMaze** | antmaze-umaze | $74.0 \pm 1.2$ | $79.1 \pm 3.4$ | $78.4 \pm 4.8$ | $87.3 \pm 2.6$ | **96.9 ± 0.8** |
> | | antmaze-umaze-diverse | $64.0 \pm 2.0$ | $66.2 \pm 1.5$ | $72.3 \pm 3.6$ | $82.1 \pm 3.3$ | **90.1 ± 0.7** |
> | | antmaze-medium-play | $68.2 \pm 3.1$ | $72.3 \pm 2.5$ | $69.5 \pm 2.8$ | $78.4 \pm 2.1$ | **82.7 ± 1.4** |
> | | antmaze-medium-diverse | $53.7 \pm 4.5$ | $57.1 \pm 2.2$ | $58.2 \pm 6.8$ | $65.2 \pm 4.5$ | **67.3 ± 2.5** |
> | | antmaze-large-play | $35.8 \pm 2.2$ | $42.8 \pm 1.5$ | $40.2 \pm 4.3$ | $51.7 \pm 4.0$ | **59.3 ± 1.5** |
> | | antmaze-large-diverse | $24.9 \pm 1.8$ | $28.1 \pm 1.7$ | $27.5 \pm 3.1$ | $36.7 \pm 2.6$ | **44.2 ± 2.7** |
> | **Maze2d** | maze2d-umaze | $72.1 \pm 6.0$ | $89.2 \pm 4.8$ | $86.6 \pm 8.1$ | $100.2 \pm 6.7$ | **113.2 ± 5.6** |
> | | maze2d-medium | $42.3 \pm 9.9$ | $63.7 \pm 8.5$ | $66.3 \pm 12.1$ | $90.8 \pm 11.5$ | **108.7 ± 4.9** |
> | | maze2d-large | $16.3 \pm 7.3$ | $55.9 \pm 9.1$ | $41.5 \pm 15.8$ | $89.2 \pm 12.2$ | **100.1 ± 7.2** |

---

> ### Author Response · Authors · 2025-11-24
>
> **W3,Q2,Q3:** The baselines compared in Table 1 and Table 2 are inconsistent: (1) Table 1 compares KATS+BC and KFC+BC, while Table 2 compares KATS+BC and KFC+CQL. Since the base algorithms of KATS+BC and KFC+CQL are different, the comparison lacks fairness. (2) In Table 1, the data augmentation methods compared are SRA, MILO, and KFC+BC, while in Table 2, the compared methods are TELS, DOGE, POR, and KFC+CQL.
>
> **Response:**
>
> We thank the reviewer for this suggestion. To address the fairness concern, we added KFC+BC to the low-data experiments in Table 2 - with the same setup as Table 2, i.e., using only 10% of the original dataset to seed data augmentation. The results (shown below) confirm that KATS+BC consistently outperforms KFC+BC in this regime.
>
> | Task | BC | KFC+BC | **KATS+BC (Ours)** |
> | :--- | :---: | :---: | :---: |
> | Antmaze-u-d | 47.1$\pm$19.8 | 53.2$\pm$9.0 | **67.5$\pm$10.3** |
> | Antmaze-u | 62.3$\pm$27.1 | 65.7$\pm$8.3 | **83.1$\pm$7.2** |
> | Antmaze-m-d | 36.7$\pm$20.5 | 43.6$\pm$11.1 | **60.3$\pm$13.8** |
> | Antmaze-m-p | 47.5$\pm$16.9 | 62.1$\pm$10.8 | **71.3$\pm$15.2** |
> | Antmaze-l-d | 25.1$\pm$12.6 | 27.0$\pm$12.2 | **42.8$\pm$11.9** |
> | Antmaze-l-p | 40.1$\pm$15.7 | 46.7$\pm$9.6 | **51.4$\pm$13.1** |
>
> We have also clarified that the two tables serve distinct purposes:
>
> *   **Table 1 (Reward-Free IL):** We compare KATS against MILO (generative adversarial IL) and SRA, a method that employs inverse dynamics to project out-of-distribution data back to the in-distribution manifold. This provides an apples-to-apples comparison of augmentation strategies in a reward-free setting.
>
>
> *   **Table 2 (Reward-Based RL):** Demonstrates that KATS can compete with reward-guided RL methods in low-data settings. We compare against classic RL algorithms (CQL, IQL) and dynamics-aware methods, specifically POR (a state-value guided approach) and TELS (which utilizes time-reversible dynamics models for regularization), to show that our trajectory-level augmentation provides sufficient signal to match methods that explicitly use rewards.
>
>
> We have included the new KFC+BC data and this experimental rationale in the revised paper to demonstrate the robustness of KATS across both imitation and reinforcement learning benchmarks.
>
> If you have any further questions, we are happy to continue the discussion:)

---

### Author Response · Authors · 2025-11-25

We sincerely thank the Reviewers (**cuKL, f7Wk, AWKq, qXWS**) for their constructive feedback. We have made some important clarifications and conducted additional experiments to address your concerns. Below is a summary of our responses indexed by topic:

### 1. Methodological Novelty & Distinction from Prior Art (KFC)
*   **KFC vs. KATS Distinction [cuKL-W1, f7Wk-Q3, f7Wk-Q4]:** We clarified that KATS enables **trajectory-level synthesis** via a closed-loop, action-independent operator, whereas KFC is mathematically restricted to single-step augmentation due to action dependence.
*   **Scalability [f7Wk-Q1]:** We explained that KATS decouples the Koopman operator from action dimensionality, resolving the parameter explosion bottleneck found in KFC.

### 2. Enhanced Experimental Rigor (Baselines & Ablations)
*   **New Ablation Studies [cuKL-W2, f7Wk-Q11]:** We added a comprehensive ablation study (analyzing IDM, learned $\sigma_\theta$ vs. analytical $\sigma_A$) to verify the necessity of each component.
*   **New Baselines [cuKL-W3, f7Wk-Q7, qXWS-W2, qXWS-W3]:** We added **KFC+BC** (for direct comparison) and **Diffuser** (state-of-the-art generative baseline), demonstrating that KATS maintains superior performance.
*   **Task Scope [AWKq-W2]:** We clarified our focus on standard MuJoCo benchmarks for fair comparison while acknowledging pixel-based domains as a future direction.

### 3. Theoretical Grounding & Applicability
*   **Symmetry Assumption [qXWS-W1]:** We provided a formal derivation showing that the closed-loop commutation property holds provided the underlying physical dynamics and expert policy are equivariant.
*   **Linearity & Deep Learning [f7Wk-Q2]:** We justified the linear operator assumption as a form of "Structured Deep Learning" that uses deep encoders to search for linearizing spaces, adhering to the "Bitter Lesson."
*   **RL Bias [AWKq-W1]:** We clarified that our method focuses on reward-free Imitation Learning, avoiding bias issues associated with action-dependent rewards in RL.

### 4. Implementation Details & Clarifications
*   **Training Pipeline [cuKL-Q1]:** We clarified that the training is **sequential**: dynamics (Eqs. 7-8) are learned first, followed by symmetry discovery (Eqs. 9-10).
*   **Adaptive Weighting [f7Wk-Q10, f7Wk-W3]:** We provided visualizations showing our adaptive weighting scheme aggressively augments in data-dense regions while remaining conservative in sparse/uncertain regions.
*   **IDM Design [f7Wk-Q9]:** We explained that the Inverse Dynamics Model (IDM) operates on raw states (rather than latent states) to avoid noise and ensure high-fidelity action inference.

If you have any further questions, we are happy to continue the discussion:)

---

### Meta-Review · Area_Chair_kdEJ · 2026-01-04

**Summary:**

Across the four reviews, the overall assessment is positive on soundness and motivation: all reviewers describe KATS as a Koopman-theory-based trajectory augmentation method for offline imitation learning that produces dynamically consistent trajectories and yields strong empirical results on standard MuJoCo/D4RL tasks when combined with behavioral cloning (BC). The main concerns informing the decision centered on (i) clarity of the method and its novelty relative to KFC (especially how trajectory-level synthesis is achieved beyond learning from transitions), (ii) experimental rigor and fairness of comparisons (missing ablations and inconsistent baselines across tables), (iii) adequacy/recency of baselines (requests for more recent trajectory generation baselines such as diffusion-based methods), and (iv) scope and assumptions underlying the closed-loop symmetry/commutation argument (how general the equivariance assumptions are, and what domains the guarantees plausibly cover). The rebuttal provides methodological clarifications and adds multiple experiments (ablation studies, additional baselines including KFC+BC in the low-data setting, and a diffusion-based baseline), which address several decision-critical points, though some presentation/notation polish and scope limitations seem to remain.

**Reviewer Concerns:**

Concerns addressed by the rebuttal:
1. Novelty vs. KFC / trajectory synthesis: Clarified that KATS enables trajectory-level synthesis via closed-loop dynamics and a commutation property, whereas KFC is limited to single-step augmentation due to action dependence; also clarified improved scalability by decoupling the operator from action dimensionality.
2. Training pipeline (Eqs. 7–8 vs. 9–10): Explained the sequential procedure: learn dynamics first (7–8), then freeze and learn symmetries (9–10).
3. Experimental rigor: Added Maze2d/AntMaze ablations (remove IDM, replace learned symmetry-weighting with analytical alternative, use KFC-style action-coupled encoder) and reported the full method performs best, attributing drops to removing IDM and losing adaptivity.
4. Fairness/inconsistent baselines: Added KFC+BC under the Table 2 low-data (10%) setting and showed KATS+BC > KFC+BC on multiple AntMaze tasks; clarified Table 1 vs. Table 2 target reward-free IL vs. reward-based offline RL, hence different comparison sets.
5. Recency of baselines: Added Diffuser and reported KATS+BC outperforms it across several low-data locomotion and AntMaze tasks.
6. Presentation clarifications: Explained “symmetry basis” as an ensemble of symmetry operators; “Sigma 0.2/0.3” as interpolation strengths; acknowledged and softened the “by construction” behavioral-consistency claim.
7. Adaptive weighting evidence: Provided Maze2d diagnostics (PCA visualization over 5k samples) plus KL/TV measures showing non-uniform weighting consistent with conservative behavior in sparse/uncertain regions.

Concerns partially addressed or still outstanding:
1. Generality of assumptions: Provided a derivation linking commutation to dynamics and policy equivariance, but applicability remains conditional and not established beyond those settings.
2. Broader evaluation: No new experiments beyond standard state-based MuJoCo; pixel-based and real-robot settings are discussed as future work.

**Reviewer Scores:**

Reviewer cuKL: Likely to increase slightly. The rebuttal addresses all main concerns: clarifies trajectory-level synthesis and the sequential training pipeline, adds the requested ablations, and resolves baseline fairness by adding KFC+BC under the Table 2 low-data setting and explaining the roles of the two tables.

Reviewer f7Wk: Likely unchanged or slightly higher. The reviewer was already positive; the rebuttal provides clear answers to clarification requests (scalability, KFC compute cost, symmetry/sigma operationalization, dataset size and sampling, adaptive weighting) and adds ablations, though the score was already high.

Reviewer AWKq: Likely unchanged. The concerns were not make-or-break and focused on broader RL settings and larger-scale evaluations; the rebuttal clarifies scope (reward-free IL) and frames these as future directions without new evidence.

Reviewer qXWS: Likely unchanged. The rebuttal addresses the main weaknesses by providing a derivation linking commutation to equivariant dynamics and policy, and by adding a diffusion-based baseline (Diffuser) with stronger low-data results for KATS+BC, though applicability remains conditional on equivariance assumptions.

---

### Decision · Program_Chairs · 2026-01-26

Accept (Poster)